# Regulation of presynaptic Ca²⁺ channel abundance at active zones through a balance of delivery and turnover

Karen L Cunningham, Chad W Sauvola, Sara Tavana, J Troy Littleton*

The Picower Institute for Learning and Memory, Department of Biology, Department of Brain and Cognitive Sciences, Massachusetts Institute of Technology, Cambridge, United States

**Abstract** Voltage-gated Ca²⁺ channels (VGCCs) mediate Ca²⁺ influx to trigger neurotransmitter release at specialized presynaptic sites termed active zones (AZs). The abundance of VGCCs at AZs regulates neurotransmitter release probability ($Pr$), a key presynaptic determinant of synaptic strength. Although biosynthesis, delivery, and recycling cooperate to establish AZ VGCC abundance, experimentally isolating these distinct regulatory processes has been difficult. Here, we describe how the AZ levels of cacophony (Cac), the sole VGCC-mediating synaptic transmission in *Drosophila*, are determined. We also analyzed the relationship between Cac, the conserved VGCC regulatory subunit α2δ, and the core AZ scaffold protein Bruchpilot (BRP) in establishing a functional AZ. We find that Cac and BRP are independently regulated at growing AZs, as Cac is dispensable for AZ formation and structural maturation, and BRP abundance is not limiting for Cac accumulation. Additionally, AZs stop accumulating Cac after an initial growth phase, whereas BRP levels continue to increase given extended developmental time. AZ Cac is also buffered against moderate increases or decreases in biosynthesis, whereas BRP lacks this buffering. To probe mechanisms that determine AZ Cac abundance, intravital FRAP and Cac photoconversion were used to separately measure delivery and turnover at individual AZs over a multi-day period. Cac delivery occurs broadly across the AZ population, correlates with AZ size, and is rate-limited by α2δ. Although Cac does not undergo significant lateral transfer between neighboring AZs over the course of development, Cac removal from AZs does occur and is promoted by new Cac delivery, generating a cap on Cac accumulation at mature AZs. Together, these findings reveal how Cac biosynthesis, synaptic delivery, and recycling set the abundance of VGCCs at individual AZs throughout synapse development and maintenance.

*For correspondence:
troy@mit.edu

## Editor's evaluation

This paper will be of interest to a broad range of neurophysiologists as it provides insights into the regulation of presynaptic voltage-gated calcium channel abundance which largely determines presynaptic strength. The findings demonstrate that while VGCC biosynthesis does not play a major role in regulating VGCC abundance at AZs, both trafficking and recycling at active zones are important regulatory steps.

## Introduction

Voltage-gated Ca²⁺ channels (VGCCs) convert electrical signals to chemical signals (Ca²⁺ influx) at the presynaptic membrane to trigger synaptic vesicle (SV) release. VGCCs are located at presynaptic active zones (AZs), specialized lipid and protein domains that cluster SVs near VGCCs and in apposition to postsynaptic neurotransmitter receptors (*Catterall and Few, 2008*; *Bucurenciu et al., 2008*;

*Eggermann et al., 2011*; *Fouquet et al., 2009*; *Kawasaki et al., 2004*; *Wang et al., 2008*; *Chen et al., 2015*; *Nakamura et al., 2015*). The strength of a synaptic connection is steeply dependent on the level of presynaptic $Ca^{2+}$ influx at AZs, highlighting that tight regulation of VGCCs is important for determining synaptic strength (*Augustine et al., 1985*; *Borst and Sakmann, 1996*; *Sheng et al., 2012*). Specifically, VGCC regulation controls neurotransmitter release probability ($P_r$; the likelihood of SV fusion following an action potential), a fundamental presynaptic component of synaptic strength. $P_r$ can vary dramatically between closely neighboring AZs, indicating local AZ mechanisms control $P_r$ (*Atwood and Karunanithi, 2002*; *Branco and Staras, 2009*; *Melom et al., 2013*; *Peled and Isacoff, 2011*; *Ariel et al., 2012*; *Akbergenova et al., 2018*; *Newman et al., 2022*). The molecular composition of the AZ represents a convergence of pathways that govern protein biosynthesis, trafficking, incorporation, and recycling of VGCCs and AZ scaffold proteins (*Petzoldt et al., 2016*). Experimentally isolating how these separate levels of regulation contribute to VGCC and AZ scaffold abundance is important for understanding how $P_r$ is established.

Multiple proteins are implicated in controlling VGCC abundance and function (*Kittel et al., 2006*; *Graf et al., 2012*; *Acuna et al., 2016*; *Zhang et al., 2016*; *Catterall and Few, 2008*; *Missler et al., 2003*; *Hoppa et al., 2012*). These regulatory pathways are of clinical importance, as disruptions in presynaptic $Ca^{2+}$ influx are linked to several diseases including migraine and ataxia (*Cao and Tsien, 2005*; *Ophoff et al., 1996*; *Pietrobon and Striessnig, 2003*; *Zhuchenko et al., 1997*; *Luo et al., 2017*; *Brusich et al., 2018*). One conserved regulator of VGCC abundance at synapses is the α2δ subunit that contributes to channel trafficking and is pharmacologically targeted by the widely prescribed drugs gabapentin and pregabalin (*Dolphin, 2016*; *Dickman et al., 2008*; *Hoppa et al., 2012*; *Eroglu et al., 2009*; *Saheki and Bargmann, 2009*; *Cassidy et al., 2014*; *Schöpf et al., 2021*). Despite the importance of α2δ and other VGCC regulatory pathways, the dynamics of channel trafficking to and from the presynaptic membrane are largely unknown due to a lack of tools for experimentally parsing the contributions of delivery versus recycling. For example, experiments in which mutant channels appear to displace wildtype channels in competition for AZ localization have generated a model where VGCCs compete for AZ 'slots', but whether this capacity for VGCCs is regulated through limited delivery or protein turnover is unknown (*Cao et al., 2004*). Indeed, the resident time of VGCCs at the presynaptic AZ membrane, to what extent VGCCs are stabilized against inter-AZ lateral transfer, and how VGCC recycling and delivery rates contribute to the overall abundance of VGCCs at release sites has not been established.

The *Drosophila* neuromuscular junction (NMJ) is an attractive system to interrogate the dynamics of VGCCs in vivo as many core proteins and pathways are evolutionarily conserved, genetic toolkits are abundant, and individual AZs are resolvable using light microscopy (*Harris and Littleton, 2015*). Additionally, while mammals have seven high voltage-activated VGCCs and four α2δ family members, evoked synaptic transmission in *Drosophila* is mediated solely by the cacophony (Cac) α1 subunit, a $Ca_v2$ channel that requires only one α2δ subunit (straightjacket) for presynaptic localization and function (*Catterall and Few, 2008*; *Kawasaki et al., 2004*; *Ryglewski et al., 2012*; *Heinrich and Ryglewski, 2020*; *Ly et al., 2008*). The *Drosophila* larval stages span roughly 6 days, during which time AZs are continuously added to the NMJ. AZs mature over a multi-day period as Cac channels and AZ scaffolds like Bruchpilot (BRP) accumulate, generating a population of AZs at multiple stages of maturation (*Fouquet et al., 2009*; *Rasse et al., 2005*; *Akbergenova et al., 2018*). This developmental heterogeneity produces functional diversity. Optical $P_r$ mapping reveals >40-fold variation in $P_r$ across the AZ population of a single motoneuron, with each NMJ housing a small population of high releasing sites among a majority of low $P_r$ AZs (*Melom et al., 2013*; *Akbergenova et al., 2018*; *Gratz et al., 2019*; *Newman et al., 2017*; *Peled et al., 2014*; *Peled and Isacoff, 2011*; *Newman et al., 2022*). $P_r$ at individual AZs correlates well with Cac abundance, demonstrating AZ Cac levels tightly regulate presynaptic output (*Akbergenova et al., 2018*; *Gratz et al., 2019*; *Newman et al., 2022*).

Here, we probe the relationship between Cac and the AZ scaffold and characterize how biosynthesis, delivery, and turnover contribute to establishing Cac abundance at AZs of the *Drosophila* NMJ. We find that AZ abundance of Cac and BRP are regulated independently at growing AZs. The AZ scaffold forms and matures in the absence of Cac channels, and BRP scaffold abundance is not a limiting factor in AZ Cac accumulation. Additionally, Cac AZ abundance is regulated downstream of a surplus of biosynthesized channels, whereas BRP biosynthesis directly reflects synaptic BRP levels. To characterize how AZ Cac is regulated downstream of biosynthesis, we used photoconversion and

photobleaching experiments in intact animals to visualize Cac delivery and recycling. Cac delivery occurs at a majority of AZs over 24 hr, correlates highly with AZ size, and is rate-limited by α2δ. Although Cac does not undergo detectable lateral transfer between neighboring AZs over the course of development, Cac recycling from the AZ membrane contributes significantly to regulating Cac AZ abundance, with new delivery driving channel turnover. Together, these data elucidate the behavior of bulk presynaptic VGCC flow across the AZ population of a single neuron.

## Results

### Ca²⁺ channels are not required for AZ scaffold formation

The AZ scaffold is required for proper clustering and accumulation of Cac channels at the *Drosophila* NMJ, as null mutants of AZ building blocks including BRP and RIM-binding protein (RBP) dramatically reduce AZ Cac abundance (*Fouquet et al., 2009*; *Liu et al., 2011*). Whether Cac channels are also required for efficient AZ scaffold formation is unclear due to the embryonic lethality of *cac* null mutants. The Flpstop conditional gene knockout approach allows single gene expression to be switched off in a cell-type specific manner by expressing a Flippase to reverse the genomic orientation of a 'Flp' construct residing in a gene of interest. A successful Flp event initiates sustained transcription of a cytosolic tdTomato reporter (*Fisher et al., 2017*). To circumvent *cac* embryonic lethality and determine whether AZs form and mature in *cac* null neurons in vivo, we expressed UAS-Flippase using a single-neuron GAL4 driver (MN1-Ib-GAL4; expressed in the Ib motoneuron innervating muscle 1) in animals carrying a Flpstop insertion in the *cac* locus (*cac*^*flp*^). Because *cac* is on the X chromosome, the presence of the tdTomato reporter in the MN1-Ib neuron of a male represents loss of the sole copy of *cac* from that neuron. To determine the timecourse of the flip event during development, we assayed the onset of expression of the tdTomato reporter. At the 1st instar stage, ~11 of the 14 (79%) MN1-Ib motoneurons in abdominal segments A1-A7 had undergone a flip event at the *cac* locus, and by the 2nd instar stage 91% were tdTomato positive (*Figure 1—figure supplement 1A, B*). Given 3rd instar MN1-Ib NMJs have fivefold more AZs than 2nd instar NMJs (*Figure 1—figure supplement 1C, D*), the majority of AZs present at the 3rd instar stage formed after the flip event occurred.

MN1-Ib synapses were divided into four groups based on the presence or absence of a co-innervating Is motoneuron, and on the presence or absence of a successful Cac-flip event (*Figure 1A*). MN1-Is and MN1-Ib identity was determined using HRP staining to mark neuronal membranes, and Cac-flip events were reported by tdTomato fluorescence. Two-electrode voltage-clamp (TEVC) recordings from muscle 1 demonstrated a dramatic reduction in EJC amplitude following stimulation in neurons that underwent a successful Cac-flip event, reflecting reduction in Cac abundance at the synaptic terminal. In Ib/Is co-innervated muscles, Cac-flipped animals had a lower EJC amplitude compared to co-innervated controls, but still supported some release due to the contribution from unaffected Is synapses. However, a successful flip event in muscles solely innervated by a Ib neuron dramatically decreased evoked responses, demonstrating Cac is not abundantly present at Cac-flipped AZs (*Figure 1B and C*). Immunostains to determine synaptic morphology (anti-HRP) and AZ number (anti-BRP) revealed these parameters are unaltered in Cac^Ib-Flp^ neurons (*Figure 1D and E*). The apposition of presynaptic AZs to postsynaptic glutamate receptor fields was also unaffected in Cac-flipped neurons (*Figure 1—figure supplement 1I-K*). Only minor alterations in terminal morphology were observed, with ectopic filopodia forming from synaptic boutons in a subset of Cac-flipped NMJs, consistent with previously reported defects in synaptically silenced *Drosophila* synapses (*Aponte-Santiago et al., 2020*; *Figure 1F*). Thus, despite the dramatic reduction in evoked release reflecting a Cac-flip event early in synapse development, AZ seeding and subsequent synaptic growth occur at a normal rate.

Although AZ seeding occurs normally without Cac, structural maturation of the AZ scaffold or accumulation of SVs could be dependent on Cac presence. Quantitative immunostaining for two AZ scaffolds demonstrated normal BRP levels and a mild increase in RBP abundance at Cac-negative AZs (*Figure 1G–I*), indicating that AZ structural maturation through the developmental acquisition of key scaffold proteins can occur independent of Cac. To further assess if AZs formed after the flip event mature at a normal rate, the AZ population was divided into two groups based on BRP abundance. BRP enrichment correlates highly with AZ age (*Akbergenova et al., 2018*), indicating that AZs with the lowest BRP enrichment represent more recently formed AZs. Both groups of the AZ population

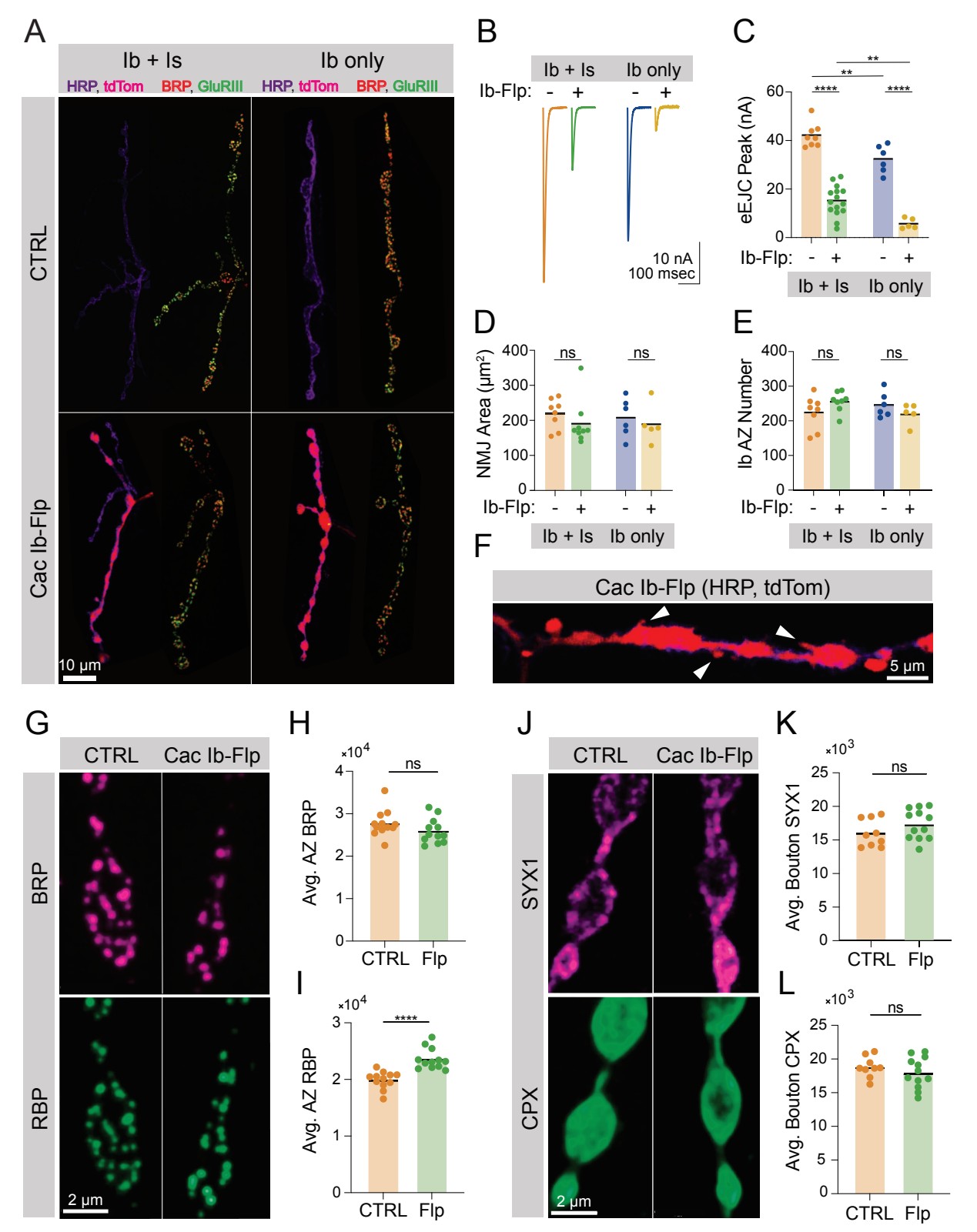

**Figure 1.** Ca²⁺ channels are not required for active zone (AZ) seeding or scaffold accumulation. (**A**) Representative images of control neuromuscular junctions (NMJs) with both Ib and Is motoneurons co-innervating muscle 1 (top left), or muscle 1 with only Ib innervation (top right), and Is and *cac*-flipped Ib neurons co-innervating muscle 1 (bottom left) or *cac*-flipped Ib neurons with no Is present (bottom right). The flip-reporter tdTomato (magenta) and HRP (neuronal membranes; purple) were imaged prior to fixation (left panels). After fixation, NMJs were immunostained for Bruchpilot

*Figure 1 continued on next page*

*Figure 1 continued*

(BRP) (red) and GluRIII (green) to label AZs and PSDs, respectively (right panels). (**B, C**) Average traces and evoked amplitudes (peak current, nA) from the indicated genotypes (co-innervated controls: orange, 42.11±1.757, n=8 NMJs from 6 animals; co-innervated Cac$^{lb-flp}$: green, 15.13±1.607, n=15 NMJs from 12 animals, p<0.0001; Ib-only controls: blue, 32.32±2.329, n=6 NMJs from 6 animals; Ib-only Cac$^{lb-flp}$: gold, 5.749±0.9544, n=5 NMJs from 5 animals, p<0.0001). (**D**) Neuronal (HRP-stained) area of Ib synapses (co-innervated controls: orange, 219.1±15.15, n=8 NMJs from 6 animals; co-innervated Cac$^{lb-flp}$: green, 190.4±21.12, 9 NMJs from 8 animals; Ib-only controls: blue, 207.8±22.4, n=6 NMJs from 6 animals; Ib-only Cac$^{lb-flp}$: gold, 189±24.59, n=5 NMJs from 5 animals). (**E**) Number of AZs formed by Ib synapses (co-innervated controls: orange, 224.3±16.88; co-innervated Cac$^{lb-flp}$: green, 255.1±10.13; Ib-only controls: blue, 245.8±14.73; Ib-only Cac$^{lb-flp}$: gold, 220.4±13.47). (**F**) Some NMJs lacking Cac display ectopic filopodia extending from boutons (white arrowheads). (**G**) Representative images of *cac*-flipped NMJs stained with anti-BRP (magenta) and anti-RIM-binding protein (RBP) (green). (**H**) Quantification of BRP abundance. Each point represents the average fluorescence intensity across the AZ population of a single NMJ (CTRL: orange, 27,736±900.2, n=12 NMJs from 6 animals; Cac$^{lb-flp}$: green, 25,943±843.7, n=12 NMJs from 6 animals). (**I**) Quantification of RBP abundance (CTRL: orange, 19,910±439.2; Cac$^{lb-flp}$: green, 23,593±530.6, p<0.0001). (**J**) Representative images of *cac*-flipped animals stained for syntaxin 1 (SYX1) (magenta) and complexin (CPX) (green). (**K**) Quantification of SYX1 abundance. Each point represents the average fluorescence intensity across the bouton population of a single NMJ (CTRL: orange, 16,028±683.5, n=9 NMJs from 5 animals; Cac$^{lb-flp}$: green, 17365±643.4, n=12 NMJs from 6 animals). (**L**) Quantification of CPX abundance (CTRL: orange, 18,810±509.4; Cac$^{lb-flp}$: green, 17,961±663.5).

The online version of this article includes the following source data and figure supplement(s) for figure 1:

**Source data 1.** Source data for *Figure 1*.

**Figure supplement 1.** Characterization of *Cac*-flipping.

**Figure supplement 1—source data 1.** Source data for *Figure 1—figure supplement 1*.

had normal BRP abundance in Cac-flipped NMJs when compared to controls, further indicating that AZ Cac presence is not required for BRP to accumulate normally (*Figure 1—figure supplement 1 E, F*). When the AZ population was divided into two groups based on RBP enrichment, both the low-RBP and high-RBP halves of the AZ population displayed increased RBP enrichment in Cac-flipped NMJs (*Figure 1—figure supplement 1G, H*). Similarly, mean bouton levels of the SV-associated protein complexin (CPX) and the t-SNARE syntaxin 1 (SYX1) were unchanged in Cac-negative neurons (*Figure 1J–L*). These data indicate that Cac abundance is critical for determining AZ function, but its presence at an AZ does not play a significant role in facilitating that AZ's formation or structural maturation, similar to observations at mammalian central nervous system (CNS) synapses (*Held et al., 2020*).

## BRP and Cac display distinct AZ accumulation trajectories

The independence of BRP scaffold formation from Cac suggests that their accumulation at growing AZs may be regulated through separate pathways. Alternatively, Cac and BRP could be co-regulated by a joint upstream component that implements a stoichiometric accumulation pattern during AZ development. It is unclear whether AZs cease accumulating new presynaptic components, or whether proteins such as BRP and Cac continue to accrue at AZs without a maturation endpoint. In a joint regulation model, the presence or absence of a cap on material accumulation would likely be the same for both proteins. Alternatively, a model where the two proteins are separately regulated would allow Cac and BRP to diverge in their accumulation trajectories over development. In wildtype larvae raised at 20°C, the larval stages are terminated after ~6 days by pupation, limiting the time available to observe AZ maturation. However, RNAi knockdown of the PTTH receptor, Torso, with a prothoracic gland GAL4 driver (*phm*-GAL4) extends larval development and triples the 3rd instar period (*Miller et al., 2012*). Quantitative imaging was used to determine whether the accumulation patterns of endogenously CRISPR-tagged Cac-GFP (*Gratz et al., 2019*) and immunostained BRP differed in long-lived larvae at day 6 and day 16 of development. These experiments test whether maturation trajectories of BRP and Cac display an endpoint, and whether they proceed stoichiometrically or diverge across extended developmental time.

Immunostaining of neuronal membranes and BRP-positive AZs demonstrated both NMJ area and AZ number increased between day 6 and day 16 (*Figure 2A–C*). These results are consistent with prior studies demonstrating muscle surface area and axonal arbors continue to grow during the extended larval period (*Miller et al., 2012*). TEVC recordings from muscle 6 revealed increased EJC amplitude in long-lived larvae at day 16 compared to day 6 (*Figure 2D and F*). This change reflects both an increase in quantal size (*Figure 2E and G*) and quantal content (*Figure 2H*). In addition, 16-day NMJs showed elevated spontaneous release frequency, consistent with the elevation in AZ number

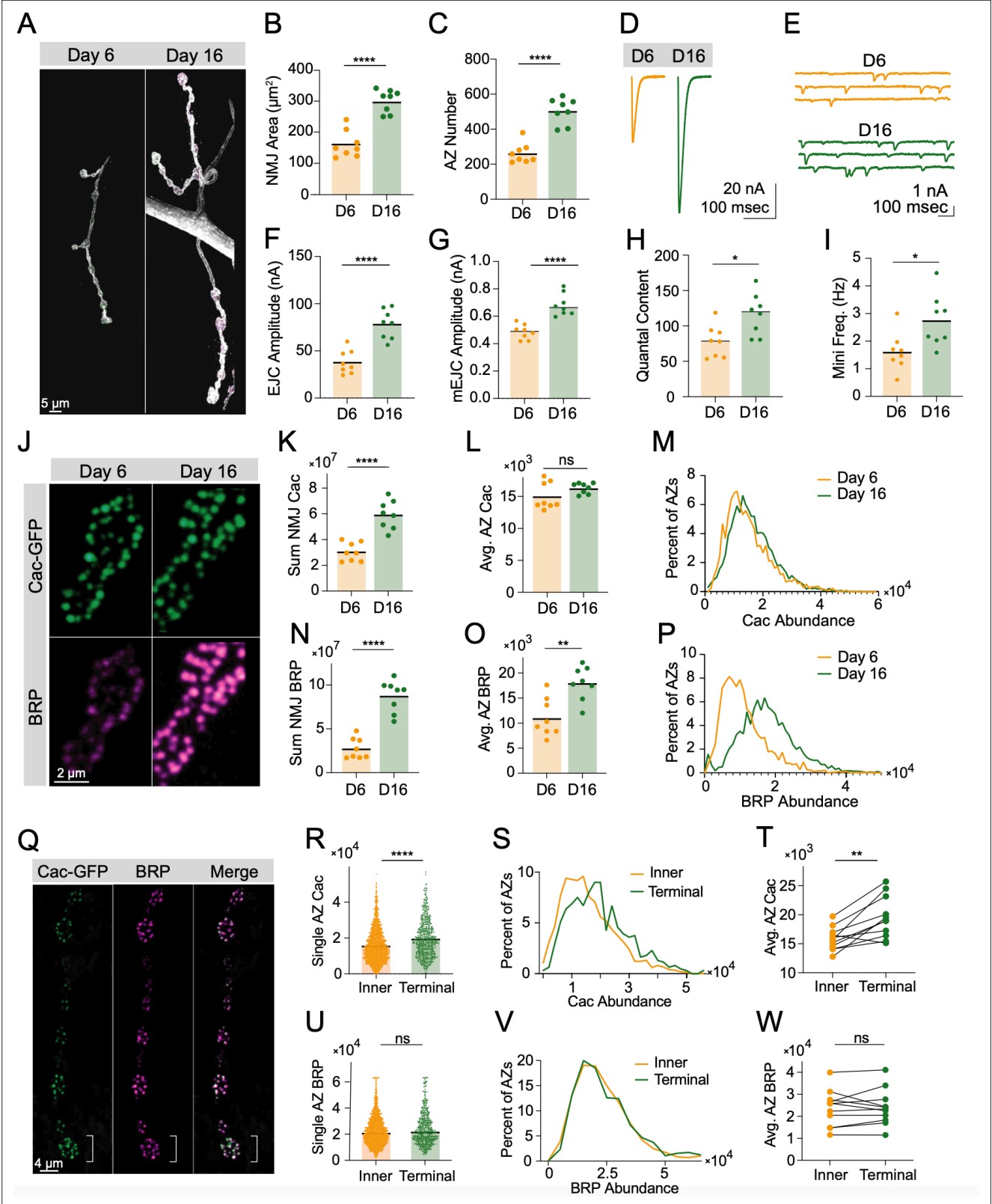

**Figure 2.** Bruchpilot (BRP) and Cac display distinct active zone (AZ) accumulation trajectories. (**A**) Representative muscle 4 neuromuscular junctions (NMJs) (stained with anti-HRP) in long-lived larvae (*phm-GAL4 >UAS-torso-RNAi*) at 6 and 16 days of larval life. (**B**) Synaptic (HRP stained) area at muscle 4 NMJs on day 6 and day 16 (D6: orange, 169.5±16.03, n=8 NMJs from 6 animals; D16: green, 316.9±13.64, n=8 NMJs from 6 animals, p<0.0001). (**C**) BRP-positive AZ number at muscle 4 NMJs on day 6 and day 16 (D6: orange, 261.6±20.02, D16: green, 504.8±25.86, p<0.0001). (**D, E**) Average

*Figure 2 continued on next page*

*Figure 2 continued*

EJC traces and representative mEJC traces at muscle 6 NMJs on day 6 and day 16. (**F**) Average evoked amplitudes (peak current, nA) (D6: orange, 38.27±4.421, n=8 NMJs from 6 animals; D16: green, 78.35±5.476, n=8 NMJs from 6 animals, p<0.0001). (**G**) Average mEJC peak amplitude (nA) (D6: 0.4862±0.01891; D16: 0.6843±0.9284, p<0.0001). (**H**) Quantal content (D6: 78.41±8.012; D16: 116.4±10.35, p=0.0116). (**I**) Average mini frequency (D6: 1.617±0.2448, D16: 2.748±0.3335, p=0.0161). (**J**) Representative synaptic boutons with endogenously tagged Cac-GFP (green) and BRP (magenta). (**K**) Sum Cac abundance per NMJ (D6: 30,486,042±2,547,718, n=8 NMJs from 6 animals; D16: 59,296,888±3,909,993, n=8 NMJs from 6 animals, p<0.0001). (**L**) Average AZ Cac-GFP signal intensity at NMJs on day 6 and day 16 (D6: 15,058±746.1; D16: 16,296±290.9). (**M**) Histogram showing distribution of Cac-GFP intensities across the AZ population at day 6 and day 16 NMJs. (**N**) Sum BRP abundance per NMJ (D6: 27,664,071±4,206,865; D16: 87,502,172±6,434,001, p<0.0001). (**O**) Average AZ BRP abundance per NMJ (D6: 11,132±1350; D16: 17,985±1178, p<0.01). (**P**) Histogram showing distribution of BRP intensities across the AZ population at day 6 and day 16 NMJs. (**Q**) Representative images of muscle 4 Ib NMJs showing Cac-GFP (green) and BRP (magenta). The white bracket marks the terminal bouton. (**R**) AZ Cac abundance at inner boutons versus terminal boutons, with each point representing the maximum fluorescent pixel intensity for one AZ (Inner: 15,879±213, n=1951 AZs; Terminal: 19,617±415.1, n=612 AZs, p<0.0001). (**S**) Distribution of AZ Cac-GFP intensities across the AZ population in inner boutons versus terminal boutons. (**T**) Average AZ Cac intensity for inner boutons versus terminal boutons. Inner boutons and terminal boutons from a single NMJ are connected with a line. (**U**) BRP abundance at AZs of inner versus terminal boutons (Inner: 23,009±262.8, n=1917 AZs; Terminal: 24,210±504.1, n=602 AZs). (**V, W**) BRP intensity distributions and pairwise comparisons of inner versus terminal boutons.

The online version of this article includes the following source data for figure 2:

**Source data 1.** Source data for *Figure 2*.

(*Figure 2E,I*). Quantitative confocal imaging of endogenously tagged Cac-GFP and immunostained BRP at long-lived NMJs confirmed the continued biosynthesis, trafficking, and delivery of these AZ components throughout development, as the sum of both Cac and BRP across the entire NMJ is elevated at day 16. However, while total BRP abundance increased by roughly fourfold at the NMJ, Cac abundance showed only an approximately twofold increase, similar to the twofold increase in AZ number (*Figure 2C, J, K and N*). Measurements of Cac-GFP at individual AZs revealed the average AZ Cac abundance in 16-day-old NMJs was not increased beyond that observed at day 6, and the distribution of Cac abundance across the AZ population did not change (*Figure 2L and M*). Thus, doubling of total Cac protein at the NMJ precisely reflected the amount necessary to maintain the AZ Cac distribution given the increase in AZ number. In contrast, BRP accumulation continued at AZs throughout the extended larval stage, resulting in a shift in the entire AZ population toward higher BRP abundance (*Figure 2O and P*), consistent with prior observations (*Perry et al., 2020*). These data demonstrate that prolonged access to new Cac and BRP results in an increased BRP-enriched AZ population with no alteration in AZ Cac accumulation, suggesting that separate regulatory mechanisms control their AZ abundance. In addition, these results suggest that Cac accumulation, but not BRP, is capped at growing AZs.

To probe whether BRP and Cac accumulation at AZs are also regulated separately in a wildtype background, we analyzed the abundance of these components at AZs across the synaptic arbor (*Figure 2Q*). Cac abundance was significantly increased at AZs in terminal boutons at muscle 4 (*Figure 2R–T*). In contrast, BRP AZ abundance was not increased in terminal boutons (*Figure 2U–W*). The unique dependence of Cac accumulation on arbor location indicates accumulation of the two proteins is regulated separately at AZs in a wildtype NMJ, consistent with their distinct accumulation trajectories in long-lived larvae.

## AZ Cac accumulation is regulated downstream of Cac biosynthesis

The cap on AZ Cac abundance across prolonged development despite continued delivery of new Cac channels suggests that regulation downstream of Cac biosynthesis allows for a stable Cac distribution to be maintained under varying environmental or developmental conditions (*Figure 2*). To determine if AZ Cac abundance is bidirectionally buffered against moderate changes in biosynthesis, the effects of increasing whole-neuron Cac levels using Cac overexpression, and decreasing Cac levels using a heterozygous *cac* deficiency, were assayed. Since a C-terminal-tagged UAS-Cac-GFP[C] overexpression construct has been demonstrated to localize to AZs and support synaptic transmission (*Kawasaki et al., 2004*), an identical C-terminal tag was inserted into the endogenous Cac locus (Cac-GFP[C]) using CRISPR so protein products of the endogenous and overexpressed locus are identical. Larvae expressing Cac-GFP[C] had normal NMJ area defined by anti-HRP staining and showed a mild decrease in AZ number (*Figure 3—figure supplement 1A-C*). In contrast to the N-terminally tagged Cac-GFP[N] used in previous experiments (*Gratz et al., 2019*), total Cac fluorescence in *Cac-GFP[C]*,

*C155>UAS-Cac-GFP* NMJs can be quantitatively imaged without the confounds of different GFP identities and tagging locations.

Quantitative confocal imaging of Cac abundance at NMJs with BRP immunostaining to label AZs revealed nearly identical AZ Cac abundance following Cac overexpression compared with controls, suggesting that Cac levels at growing AZs are rate-limited downstream of Cac biosynthesis, and that Cac channels compete for limited delivery or incorporation into growing AZs (*Figure 3A–C*). Overexpression of red-tagged UAS-Cac-tdTomato in the CRISPR-tagged Cac-GFP background directly demonstrated this competition between Cac channels, as Cac-tdT overexpression resulted in an ~70% decrease in Cac-GFP abundance at AZs (*Figure 3D and E*). TEVC recordings demonstrated that Cac overexpression does not significantly increase release, consistent with the failure of Cac overexpression to drive increased Cac abundance at AZs (*Figure 3F and G*). Together, these results indicate that AZs resist over-accumulating Cac channels in conditions of increased cellular availability. This reflects observations in extended larval stage animals and suggests that AZ Cac incorporation is limited downstream of Cac biosynthesis, perhaps by regulated trafficking, incorporation, or retention at synapses.

Though Cac biosynthesis is not a limiting regulatory point of AZ Cac incorporation, it is possible that Cac biosynthesis precisely matches Cac demand in growing NMJs. In this model, moderate reductions in Cac biosynthesis should result in similar reductions in AZ Cac levels. Alternatively, Cac could be biosynthesized in excess of AZ demand in wildtype situations, resulting in AZs being buffered against moderate reductions in Cac biosynthesis. Western blot analysis of adult head extracts of female animals with endogenously tagged Cac-GFP in heterozygous combination with a chromosomal deletion spanning the *cac* locus (*cac^{GFP/Df}*) revealed that whole-brain Cac levels in *cac^{GFP/Df}* heterozygotes were reduced to 52% of control (homozygous *cac^{GFP}*) levels (*Figure 3H,I*). Despite the twofold reduction in Cac biosynthesis, MN4-Ib AZs in *cac^{GFP/Df}* heterozygotes retained 90% of control Cac-GFP levels without a change in AZ number (*Figure 3J–M*). Quantitative confocal imaging of Cac-GFP fluorescence in female larvae expressing endogenously tagged Cac-GFP in combination with untagged wildtype Cac (*cac^{GFP/+}*) demonstrated exactly 50% of AZ fluorescence intensity when compared to *cac^{GFP}* homozygotes, verifying these methods accurately report fold-change in protein abundance (*Figure 3—figure supplement 2A, B*). Other image analysis metrics, including sum GFP fluorescence in ROIs covering each AZ, revealed similarly robust quantification (*Figure 3—figure supplement 2C-G*).

While bidirectional buffering mitigates AZ Cac depletion in *cac* deficiency heterozygotes, even small changes in Cac abundance are predicted to impact release due to the steep reliance of $P_r$ on presynaptic $Ca^{2+}$ influx. Beyond the 10% decrease in AZ Cac levels at MN4-Ib terminals, Cac-GFP intensity at the co-innervating MN4-Is AZs revealed a greater decrease of 30%, perhaps due to the bigger size of the Is terminal which innervates a large subset of muscles and may have increased demand for Cac channels (*Figure 3N*). Electrophysiological recordings in *cac^{GFP/Df}* heterozygous NMJs confirmed a 60% decrease in evoked peak current compared to controls, corresponding to the composite 10% and 30% reductions in AZ Cac abundance in Ib and Is neurons, respectively (*Figure 3O and P*). This highlights the functional significance of mild alterations in AZ Cac abundance. Overall, these results demonstrate that at wildtype synapses, Cac AZ abundance is buffered against moderate decreases or increases in biosynthesis, suggesting that the rate-limiting factor for AZ Cac incorporation is situated downstream of its biosynthesis levels.

## BRP biosynthesis rate-limits BRP accumulation at AZs

In contrast to Cac, AZs continue to accrue BRP given extended developmental time (*Figure 2*), suggesting that BRP biosynthesis may rate-limit AZ BRP incorporation into growing AZs. In a biosynthesis-limiting model, moderate decreases in neuronal BRP abundance should result in comparable decreases of the protein at AZs. This hypothesis is consistent with prior observations that AZ BRP abundance is negatively correlated with AZ number in mutants with altered AZ number (*Goel et al., 2019*). Reductions in BRP abundance using either a heterozygous BRP deficiency (*brp^{Df/+}*) or pan-neuronal expression of BRP RNAi revealed a linear relationship between whole-cell and AZ BRP abundance (*Figure 4A–D*). Western blot analysis of head extracts demonstrated that *brp^{Df/+}* heterozygotes produced 65% of control levels of the protein, with AZ BRP abundance reduced to 64% of control levels. Similarly, pan-neuronal RNAi knockdown reduced BRP levels to 14% in whole brains, resulting in a comparable reduction to 15% at AZs. These results are consistent with the increase in

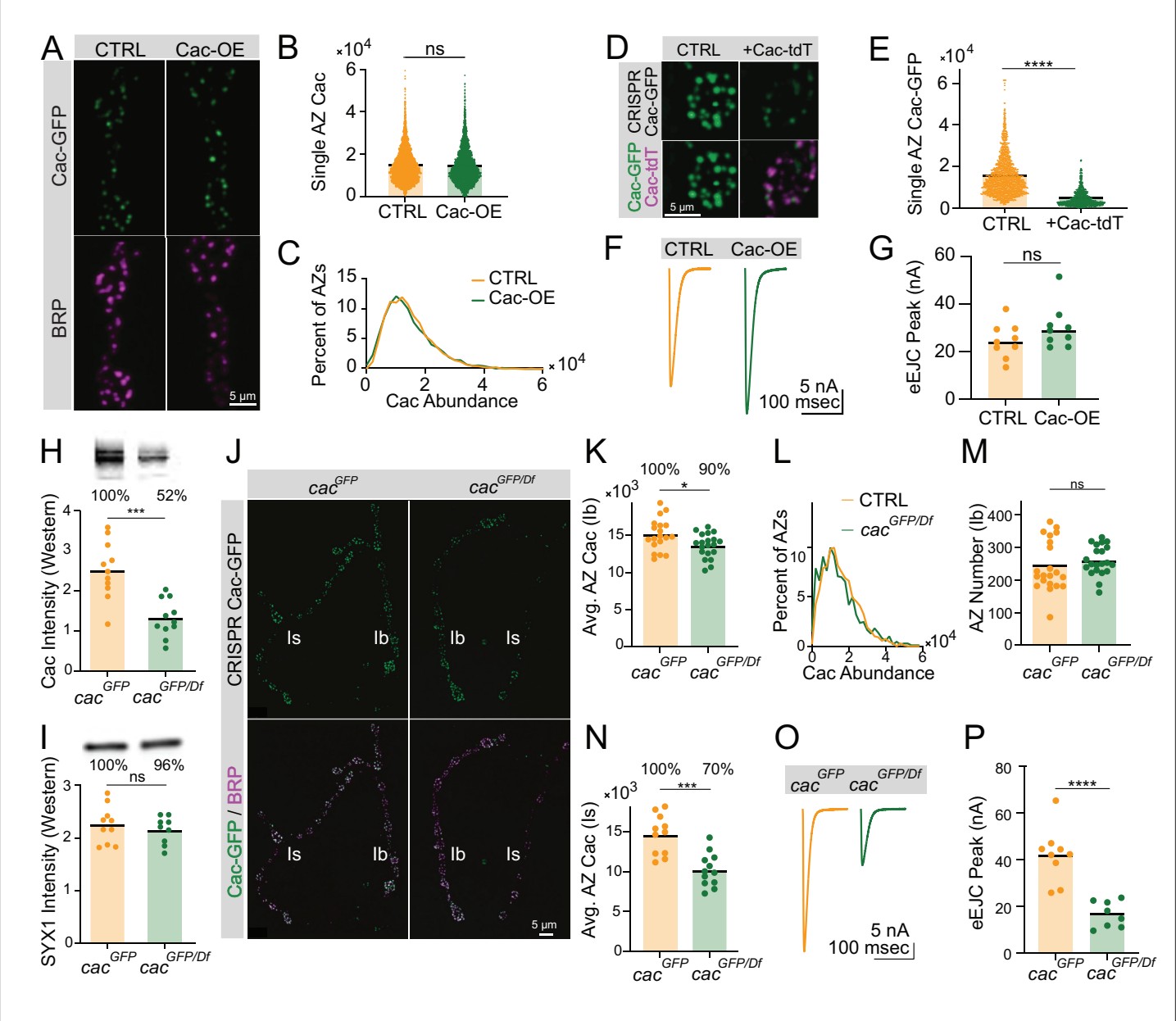

**Figure 3.** Active zone (AZ) Cac accumulation is regulated downstream of Cac biosynthesis. (**A**) Representative images of Cac-GFP (green) and Bruchpilot (BRP) (magenta) at muscle 4 for endogenously C-terminally tagged Cac-GFP controls ($cac^{GFP(C)},C155/cac^{GFP(C)}$) and Cac overexpression animals ($cac^{GFP(C)},C155/cac^{GFP(C)},UAS-Cac-GFP$). (**B**) Quantification of AZ Cac intensity in control versus Cac overexpression neuromuscular junctions (NMJs). Each point represents the maximum Cac-GFP pixel intensity of a single AZ (control: 14617±130.8, n=3663 AZs from 10 NMJs from 6 animals; Cac OE: 14,383±148.2, n=3145 AZs from 10 NMJs from 6 animals). (**C**) Distribution of AZ Cac-GFP abundance across the AZ population in control versus Cac overexpression NMJs. (**D**) Representative images of endogenously tagged Cac-GFP (green) and overexpressed Cac-tdTomato (magenta) at AZs in control ($cac^{GFP},C155$) versus Cac-tdTomato overexpression ($cac^{GFP},C155;UAS-Cac-tdT$). (**E**) Quantification of AZ Cac-GFP intensity with and without Cac-tdTomato overexpression (control: 15,339±232, n=1613 AZs, Cac-tdT: 4477±123.9, n=850 AZs, p<0.0001). (**F, G**) Average traces and evoked peak currents (nA) at muscle 6 in Cac overexpression and control animals (control: 24.45±2.433, n=9 NMJs from 5 animals; Cac OE: 30.11±3.046, n=9 NMJs from 5 animals). (**H, I**) Quantifications and representative images of Western blots of adult head extracts from $cac^{GFP/Df}$ heterozygotes and $cac^{GFP}$ controls (n=11 samples, extracts from 5 adult heads per sample). Each point in **H** represents Cac intensity in one lane ($cac^{GFP}$: 25,092±2146; $cac^{GFP/Df}$: 13,055±1403, p<0.001). Each point in **I** represents syntaxin 1 (SYX1) intensity in one lane ($cac^{GFP}$: 224,900±11,239; $cac^{GFP/Df}$: 214,889±86,96). Representative images for Cac and SYX are shown above each quantification. (**J**) Representative images of endogenous Cac-GFP (green) and BRP (magenta) at muscle 4 AZs in $cac^{GFP/Df}$ heterozygotes and $cac^{GFP}$ controls. Ib and Is terminals are labeled in white. (**K**) Quantification of average Cac-GFP AZ abundance per MN4-Ib NMJs ($cac^{GFP}$: 15,105±486.3, n=19 NMJs from 7 animals; $cac^{GFP/Df}$: 13,617±365.9, n=21 NMJs from 7 animals, p<0.05). (**L**) Histogram of Cac-GFP intensity across the AZ population in $cac^{GFP/Df}$ heterozygotes and $cac^{GFP}$ controls at MN4-Ib NMJs. (**M**) BRP-positive AZ number per MN4-Ib NMJ ($cac^{GFP}$:

*Figure 3 continued on next page*

*Figure 3 continued*

243.4±17.05; *cac^GFP/Df*: 260.3±9.963). (**N**) Quantification of average AZ Cac-GFP intensity per Is innervation of muscle 4 NMJs (*cac^GFP*: 14,621±747.5, n=11 NMJs from 7 animals; *cac^GFP/Df*: 10,219±601 n=12 NMJs from 7 animals, p<0.001). (**O, P**) Average traces and evoked peak currents (nA) at muscle 6 in *cac^GFP/Df* heterozygotes and *cac^GFP* controls (*cac^GFP*: 41.96±3.879, n=9 NMJs from 5 animals; *cac^GFP/Df*: 16.7±1.984, n=8 NMJs from 7 animals, p<0.0001).

The online version of this article includes the following source data and figure supplement(s) for figure 3:

**Source data 1.** Source data for *Figure 3*.

**Source data 2.** Western regions used for *Figure 3*.

**Source data 3.** Western for *Figure 3*.

**Figure supplement 1.** Synaptic morphology of Cac-GFP C-terminal CRISPR-tagged larvae.

**Figure supplement 1—source data 1.** Source data for *Figure 3—figure supplement 1*.

**Figure supplement 2.** Quantitative imaging control.

**Figure supplement 2—source data 1.** Source data for *Figure 3—figure supplement 2*.

AZ BRP observed during extended larval stages and indicate that BRP biosynthesis determines AZ BRP abundance.

## BRP enrichment does not limit AZ Cac accumulation

Given the requirement of BRP for proper Cac accumulation at AZs (*Fouquet et al., 2009*), it is possible that BRP abundance is a limiting factor for Cac incorporation into growing AZs. An alternative model is that Cac accumulation requires BRP but is not limited by the amount of BRP at AZs, consistent with the observation that elevated BRP levels at AZs of 16-day-old long-lived larvae do not promote increased Cac levels (*Figure 2*). Quantitative confocal imaging of endogenously tagged Cac-GFP at AZs in control, *brp^Df/+* heterozygotes, and pan-neuronal BRP knockdown revealed that BRP is required but not rate-limiting for AZ Cac accumulation (*Figure 4A and E*). BRP knockdown to 15% of control BRP levels resulted in a moderate reduction of Cac AZ abundance (69% of control levels), reflecting the importance of BRP in facilitating AZ Cac accumulation as previously reported (*Fouquet et al., 2009*). However, although AZ BRP is reduced by 36% in *brp^Df/+* heterozygotes, Cac-GFP is present at normal levels (*Figure 4B and E*). Evoked synaptic transmission following single action potentials was unaffected by this 36% reduction in AZ BRP, further indicating Cac abundance is unaffected by a moderate reduction in AZ BRP enrichment (*Figure 4F and G*).

To further assess the extent to which AZ BRP enrichment is required for normal Cac accumulation, AZs in NMJs of animals expressing pan-neuronally driven BRP RNAi were subdivided into two groups based on the presence or absence of BRP. When all Cac-positive AZs were analyzed regardless of their BRP presence or absence, AZ Cac levels were significantly lower at BRP RNAi NMJs (*Figure 4—figure supplement 1A-C*). However, when only the BRP-positive AZs were analyzed, Cac levels were normal despite an ~80% reduction in BRP abundance compared to controls (*Figure 4—figure supplement 1D-G*). These data further indicate that although BRP abundance may rate-limit other aspects of AZ structural or functional maturation, it is not a limiting factor in Cac accumulation at AZs.

## Cac accumulation at AZs is dosage sensitive to the α2δ subunit

Across multiple systems, the highly conserved VGCC subunit α2δ promotes channel surface expression. In rodents, α2δ overexpression leads to increased surface VGCC density and increased synaptic release, indicating α2δ is a rate-limiting factor in VGCC accumulation (*Hoppa et al., 2012*). In *Drosophila*, mutations in the α2δ subunit (*straightjacket*) lead to reduced Cac at nerve terminals, reduced synaptic growth, and failure of synaptic homeostasis (*Ly et al., 2008*; *Dickman et al., 2008*; *Kurshan et al., 2009*; *Wang et al., 2016*). Consistently, quantitative confocal imaging of endogenously tagged Cac-GFP in animals heterozygous for a combination of the α2δ² null allele and an α2δ splice site mutant that reduces full-length protein expression (*cac^GFP*; α2δ^2/k10814) revealed a large decrease in AZ Cac levels to 36% of control levels, with a third of AZs lacking detectable Cac (*Figure 5A–C*). TEVC recordings revealed a severe decrease in evoked current, reflecting the reduction in Cac levels at this terminal (*Figure 5D and E*). Imaging of Cac-GFP in animals heterozygous for the α2δ² null allele (*cac^GFP*;α2δ^2/+) indicates that Cac abundance at AZs is dosage sensitive to α2δ, as Cac-GFP was reduced

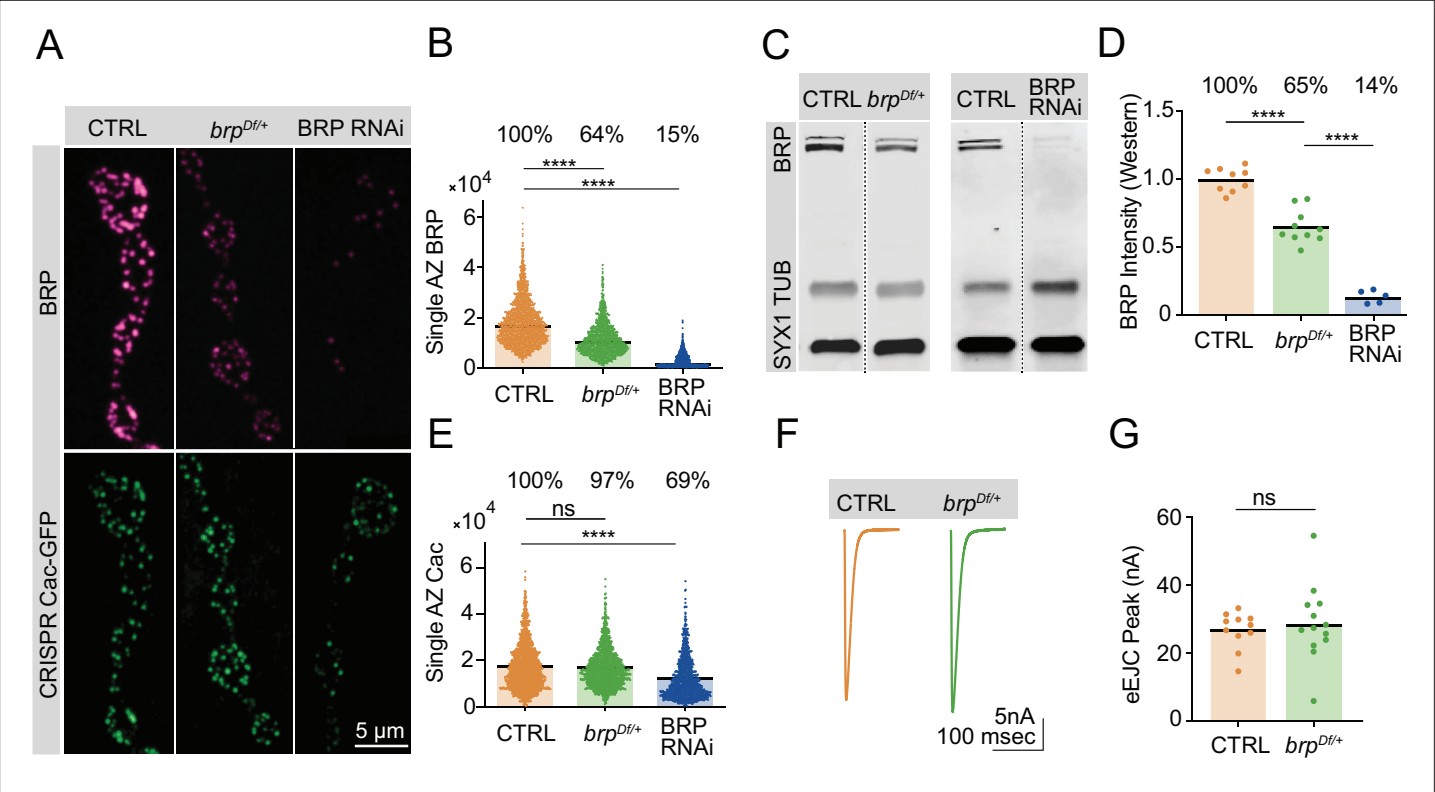

**Figure 4.** Bruchpilot (BRP) biosynthesis rate-limits active zone (AZ) BRP incorporation. (**A**) Representative images of BRP and endogenously tagged Cac-GFP at AZs in controls, *brp^{Df/+}* heterozygotes and pan-neuronally (*elav-GAL4*) expressed BRP RNAi. (**B**) Quantification of single AZ BRP intensity, with average AZ BRP intensity listed as a percent of control above each genotype (control: 18,161±210.7, n=1779 AZs; *brp^{Df/+}*: 11,534±176.7, n=1350 AZs, p<0.0001; BRP RNAi: 2806±79.91, n=1140 AZs, p<0.0001). (**C**) Representative image of Western blots of adult head extracts from control and *brp^{Df/+}* heterozygotes (left panel), and control and pan-neuronally expressed BRP RNAi animals (right panel) stained for syntaxin 1 (SYX1) (loading control), Tubulin, and BRP. (**D**) Quantification of BRP intensity in Western blots of the indicated genotypes. Each point represents BRP intensity in one lane, with BRP intensity normalized to the SYX1 loading control. Percent of protein abundance compared to control (100%) is shown above each genotype (control: 1.0±0.02912, n=9 lanes; *brp^{Df/+}*: 0.6518±0.03862, n=10 lanes, p<0.0001; BRP RNAi: 0.1366±0.02117, n=5 lanes, p<0.0001). (**E**) Quantification of endogenously tagged Cac-GFP intensity at single AZs in controls, *brp^{Df/+}* heterozygotes, and pan-neuronally expressed BRP RNAi (control: 17,268±232.4, n=1757 AZs; *brp^{Df/+}*: 16,788±217, n=1403 AZs; BRP RNAi: 11,959±259, n=1140 AZs, p<0.0001). (**F, G**) Average traces and quantified evoked peak currents (nA) in control and *brp^{Df/+}* heterozygotes at muscle 6 (control: 26.78±1.632, n=11 neuromuscular junctions (NMJs); *brp^{Df/+}*: 28.71±3.105, n=13 NMJs).

The online version of this article includes the following source data and figure supplement(s) for figure 4:

**Source data 1.** Source data for *Figure 4*.

**Source data 2.** Western for *Figure 4* panel C - BRP deficiency heterozygote.

**Source data 3.** Western regions used for *Figure 4* panel C - BRP RNAi.

**Figure supplement 1.** Single active zone (AZ) analysis of Cac and Bruchpilot (BRP) following BRP RNAi knockdown.

**Figure supplement 1—source data 1.** Source data for *Figure 4—figure supplement 1*.

to 83% of control levels in the *α2δ^{2/+}* heterozygote, with release significantly reduced compared to controls (*Figure 5A–E*).

To further examine whether AZ Cac accumulation is rate-limited by α2δ, UAS-α2δ was pan-neuronally overexpressed in the CRISPR Cac-GFP background (*cac^{GFP},elav-Gal4;;UAS-α2δ*). α2δ over-expression increased average AZ Cac abundance to 123% of driver-only control levels (*Figure 5F and G*), in contrast with Cac-GFP overexpression, which failed to promote increased Cac accumulation at AZs (*Figure 3A–C*). However, this increase in AZ Cac abundance was non-uniform across the AZ population, as AZs with the highest levels of Cac were unaffected by α2δ overexpression (*Figure 5H*). Because this population is predicted to be the high releasing AZs that drive a majority of release in low external Ca^{2+} (*Akbergenova et al., 2018*), and because low $P_r$ AZs facilitate while high $P_r$ AZs depress

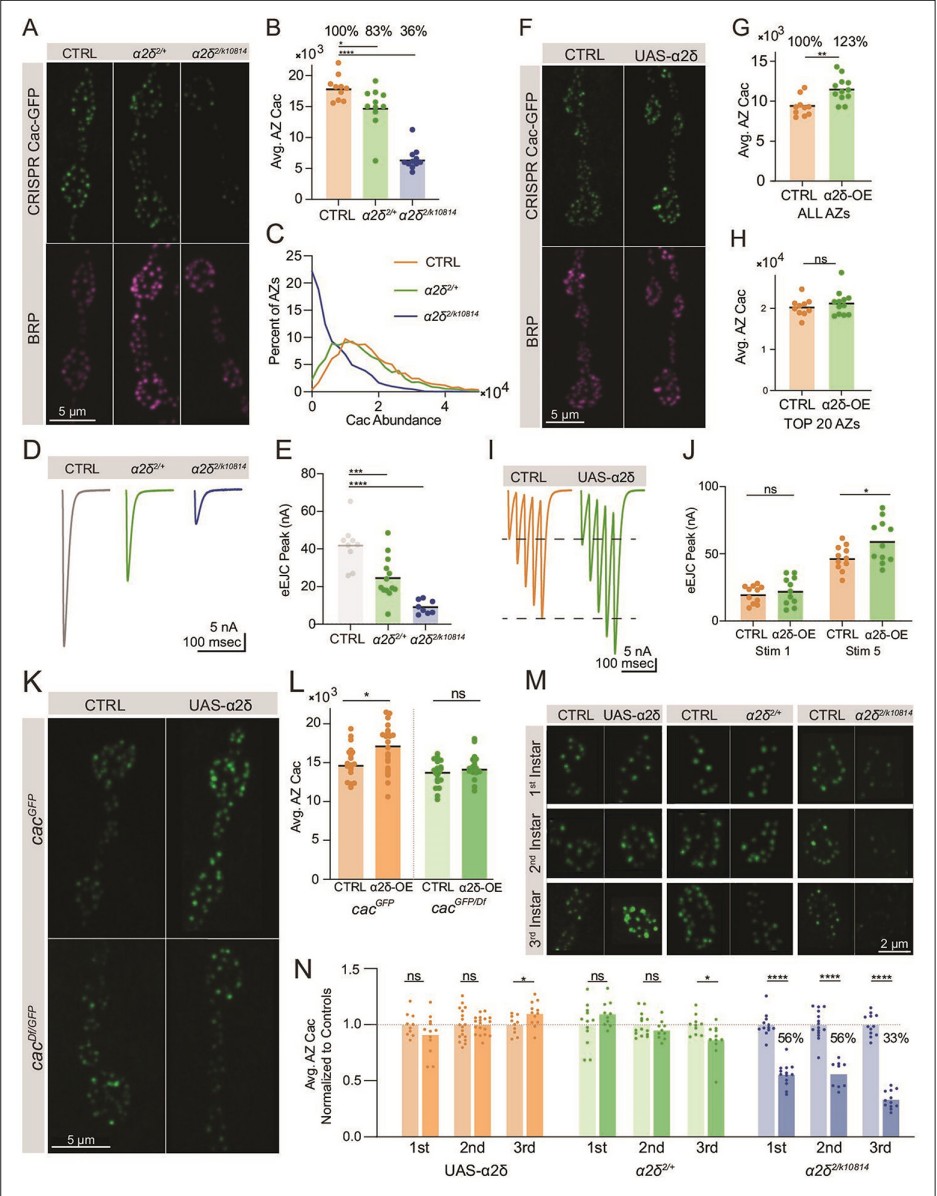

**Figure 5.** Cac accumulation at active zones (AZs) is dosage sensitive to the α2δ subunit. (**A**) Representative images of endogenously tagged Cac-GFP (green) and Bruchpilot (BRP) (magenta) in controls (*cac^GFP*), *α2δ* null heterozygotes (*cac^GFP;α2δ^2/+*) and *α2δ^2/k10814* transheterozygotes (*cac^GFP;α2δ^2/k10814*). (**B**) Quantification of average AZ Cac-GFP fluorescence of neuromuscular junctions (NMJs) in each genotype. Cac abundance as a percent of control levels is shown above each genotype (control: 18,007±641.1, n=10 NMJs from 5 animals; *α2δ^2/+*: 14,926±1014, n=11 NMJs from 6 animals, p=0.0174; *α2δ^2/k10814*: 6542±541.3, n=11 NMJs from 6 animals, p<0.0001). (**C**) Frequency distribution of single AZ Cac-GFP intensity across the AZ population in each genotype. (**D, E**) Average traces and quantified evoked peak current (nA) at muscle 6 for each genotype. Recordings were collected in combination with *cac^GFP/Df* heterozygotes (*Figure 3P*) and the control is replicated here in gray for ease of comparison (control: 41.96±3.879, n=9 NMJs; *α2δ^2/+*: 24.53±3.133, n=13 NMJs, p<0.001; *α2δ^2/k10814*: 9.161±1.268, n=8 NMJs, p<0.0001). (**F**) Representative images of endogenously tagged Cac-GFP (green) and BRP (magenta) in control (*cac^GFP,C155*) and *α2δ* overexpression animals (*cac^GFP,C155;;UAS-α2δ*). (**G**) Average AZ Cac abundance per NMJ (control: 9332±388.6, n=10 NMJs from 5 animals; *α2δ* OE: 11497±449.8, n=12 NMJs from 6 animals, p=0.0019). (**H**) Average of top 20 brightest AZs per NMJ (control: 22,996±752.5; *α2δ* OE: 24,823±1011). (**I, J**) Average traces and quantified 1st and 5th stimulus evoked peak current at muscle 6 in 0.25 mM external Ca^2+ for controls and *α2δ* overexpression animals (control 1st: 19.87±1.98, n=11 NMJs; *α2δ* OE 1st: 22.45±3.127, n=11 NMJs; control 5th: 46.82±2.794; *α2δ* OE 5th: 59.21±4.839, p=0.0249). (**K, L**) Representative images and quantification of endogenously tagged Cac-GFP with (right) and without (left) *α2δ* overexpression in either control

*Figure 5 continued on next page*

*Figure 5 continued*

(top) or *cac^{GFP/Df}* heterozygous (bottom) NMJs (*cac^{GFP}*,*C155*: 15,105±486.3, n=19 NMJs; *cac^{GFP}*,*C155;UAS-α2δ/+*: 17,142±814.4, n=15 NMJs, p<0.0225; *cac^{Df}/cac^{GFP}*,*C155*: 13,617±365.9, n=20 NMJs; *cac^{Df}/cac^{GFP}*,*C155;UAS-α2δ/+*: 14,895±587, n=12 NMJs). (**M, N**) Representative images and quantification of average AZ Cac intensity per NMJ of endogenously tagged Cac-GFP in 1st, 2nd, and 3rd instar muscle 4 Ib NMJs in α2δ overexpression, *α2δ^{2/+}*, and *α2δ^{2/k10814}* animals. Each pairwise comparison was normalized so that the control average is 1.0 (*α2δ*-OE 1st: 0.9125±0.0496, n=12 NMJs; *α2δ*-OE 2nd: 1±0.019, n=18 NMJs; *α2δ*-OE 3rd: 1.098±0,03132, n=12 NMJs, p=0.0296; *α2δ^{2/+}*1st: 1.096±0.0361, n=10 NMJs; *α2δ^{2/+}* 2nd: 0.9525±0.02574, n=10 NMJs; *α2δ^{2/+}* 3rd, p=0.0213; 0.8661±0.04408, n=11 NMJs; *α2δ^{2/k10814}* 1st: 0.5569±0.02853, n=15 NMJs, p<0.0001; *α2δ^{2/k10814}* 2nd: 0.5619±0.03902, n=9 NMJs, p<0.0001; *α2δ^{2/k10814}* 3rd: 0.3344±0.02298, n=12 NMJs, p<0.0001).

The online version of this article includes the following source data and figure supplement(s) for figure 5:

**Source data 1.** Source data for *Figure 5*.

**Figure supplement 1.** α2δ's role in regulating neuromuscular junction (NMJ) morphology is not dosage-sensitive and its Cac localization phenotype is independent of axon length.

**Figure supplement 1—source data 1.** Source data for *Figure 5—figure supplement 1*.

(*Peled and Isacoff, 2011*; *Newman et al., 2017*), the increase in Cac at low-releasing sites due to α2δ overexpression is predicted to have a limited effect on single action potential release (dominated by high $P_r$ AZs), but would enhance facilitation during a short train of action potentials (when lower $P_r$ AZs contribute more to net release). Consistent with this prediction, TEVC recordings showed that evoked peak current in the first stimulation in a train of five action potentials was unaffected by α2δ overexpression, but an increase in response was observed by the fifth stimulus (*Figure 5I and J*). Together, these results demonstrate that unlike BRP, which is required but not rate-limiting for Cac accumulation at AZs, α2δ is both required and rate-limiting for this process.

The mechanism by which α2δ rate-limits Cac AZ abundance is unclear, as α2δ could control a bottleneck in Cac trafficking and/or promote Cac stabilization at AZs' post-incorporation. At NMJs of 3rd instar larvae, AZ Cac abundance is almost fully maintained at AZs even when total neuronal Cac abundance is reduced by 50% (*Figure 3*), indicating that only a fraction of total Cac channels are mobilized for AZ occupancy, and that this fraction is modulated depending on total Cac availability. To test whether α2δ controls the fraction of biosynthesized Cac destined for AZs, we tested whether α2δ overexpression can increase AZ Cac in the *cac^{GFP/Df}* heterozygous background. While α2δ overexpression resulted in elevated AZ Cac in control animals, it did not increase AZ Cac in the *cac^{GFP/Df}* deficiency background, indicating that overexpressed α2δ relies on the presence of a non-AZ Cac pool for its ability to enhance Cac abundance at AZs (*Figure 5K and L*). These data suggest that α2δ mediates Cac trafficking rather than channel stabilization. A forward trafficking role could facilitate progression through the biosynthetic pathway or long-range axonal transport. However, the severity of the Cac-depletion phenotype in *α2δ^{2/k10814}* mutants is independent of axon length, with distal (segments 5 and 6) and proximal (segments 1 and 2) abdominal segments displaying similar reductions in Cac compared to controls (*Figure 5—figure supplement 1D, E*). These data are consistent with a role for α2δ in mediating progression through the ER/Golgi pathway prior to long-range axonal transport. Imaging of endogenously tagged Cac-GFP following α2δ overexpression or reduction in 1st, 2nd, and 3rd instar animals demonstrated that α2δ becomes rate-limiting for AZ Cac over developmental time, with α2δ overexpression and *α2δ^{2/+}* heterozygote phenotypes appearing during the 3rd instar stage when a large majority of new AZs are forming. Similarly, the *α2δ^{2/k10814}* phenotype is present across all stages of larval development but becomes more severe in the 3rd instar stage, likely due to an increased demand for Cac trafficking and delivery as AZ number increases (*Figure 5M and N*). Together, these results demonstrate that α2δ plays a rate-limiting role in AZ Cac accumulation in a regulatory step upstream of AZ Cac delivery, and that the need for normal α2δ levels increases throughout developmental time as the number of AZs increases.

Similar to prior reports, a modest decrease in synapse area and AZ number was observed in the more severe α2δ reduction (*α2δ^{2/k10814}*), reflecting a role for α2δ in morphological development of the NMJ (*Figure 5—figure supplement 1A-C*; *Kurshan et al., 2009*). The requirements of α2δ in morphological development and Cac accumulation have been shown to represent independent functions of the subunit (*Kurshan et al., 2009*). That AZs and NMJs form normally in the absence of Cac channels (*Figure 1*), but abnormally in the *α2δ^{2/k10814}* mutant, supports the model that the morphological role

of α2δ is not downstream of its role in Cac localization. Moreover, unlike α2δ's requirement in Cac abundance, the morphological phenotype is not dosage-dependent, as $α2δ^{2/+}$ heterozygotes were morphologically normal (**Figure 5—figure supplement 1**).

## Cac delivery to AZs correlates with AZ size and is regulated by α2δ

To establish a protocol to directly measure the delivery of new Cac channels, entire Ib NMJs innervating muscle 26 in late 2nd instar animals expressing endogenously tagged Cac-GFP and GluRIIA-RFP were photobleached and synapses were imaged immediately after photobleaching to confirm loss of the Cac-GFP signal (**Figure 6—figure supplement 1A**). Animals were then allowed to recover and grow for 24 hr before dissection. Fixed NMJs were imaged 24 hr post-bleach alongside unbleached NMJs as a control (**Figure 6—figure supplement 1B**). New synapse growth and new AZ addition (1.9-fold increase in AZ number) were observed after 24 hr of growth, as previously described (**Akbergenova et al., 2018**; **Figure 6—figure supplement 1C**). AZ ROIs were defined using unbleached GluRIIA-labeled PSDs. Total Cac signal inside each ROI was measured and averaged at AZs across the NMJ, demonstrating near-complete photobleaching and partial recovery of Cac-GFP signal 24 hr post-bleach, representing new Cac-GFP channels that arrived at NMJs after photobleaching (**Figure 6—figure supplement 1A, B, D**).

This FRAP protocol was used to measure Cac-GFP delivery over 24 hr in control and $α2δ^{2/k10814}$ NMJs, and recovered Cac levels were correlated with post-fixation BRP immunostaining (**Figure 6A**). Cac delivery was broadly distributed across AZs of the NMJ, rather than targeted to a specific subset of AZs. In the span of 24 hr, 90% of AZs had received detectable new Cac-GFP channels (**Figure 6B**). New Cac-GFP signal was roughly 50% dimmer than unbleached Cac-GFP control AZs, consistent with prior demonstration that AZs grow to full structural and functional maturity over several days (**Akbergenova et al., 2018**; **Figure 6C**). To determine whether new channels were delivered preferentially to AZs based on AZ size or age, 24 hr delivery of CRISPR Cac-GFP was correlated with BRP abundance, an established correlate of AZ age and functional maturity (**Akbergenova et al., 2018**). In non-photobleached NMJs, Cac-GFP correlated highly with BRP abundance at individual AZs, with an average Pearson r of 0.88, consistent with prior studies (**Figure 6D and F**; **Akbergenova et al., 2018**). After 24 hr of new channel delivery post-photobleaching, Cac-GFP signal correlated similarly with BRP at AZs, indicating that a larger number of newly delivered channels were incorporated at mature AZs over a 24 hr period, and new delivery correlated with AZ size and maturity (**Figure 6E and F**). Twenty-four hour Cac delivery correlated similarly well with GluRIIA-RFP abundance, another established proxy for AZ age (**Figure 6—figure supplement 1E-G**; **Akbergenova et al., 2018**).

Cac levels in $α2δ^{2/k10814}$ mutants in unbleached segments were reduced by roughly 70% at AZs, similar to the Cac reduction measured at muscle 4 (**Figure 6G**, **Figure 5A and B**). After 24 hr of recovery, $α2δ^{2/k10814}$ mutants showed an ~50% reduction in Cac delivery compared to controls, with no change in AZ number, suggesting that α2δ is required for Cac delivery (**Figure 6H,I**). Although reduced, residual Cac delivery remained correlated with BRP abundance in $α2δ^{2/k10814}$ mutants (**Figure 6J–L**). These data indicate that Cac delivery is regulated by α2δ, is broadly available to AZs across the NMJ, and its incorporation positively correlates with AZ size and age.

## Cac turnover contributes to setting AZ Cac abundance

The leveling-off of Cac accumulation at mature AZs despite continued delivery predicts that Cac turnover must occur. The efficiency of Cac surface retention and timecourse of VGCC removal from AZs has not been measured in any in vivo system to our knowledge. Additionally, the mechanism by which VGCC regulatory proteins promote Cac surface expression has remained unclear due to difficulties experimentally distinguishing Cac trafficking/delivery versus Cac turnover. To facilitate direct measurements of Cac removal from AZs, an endogenously tagged photoconvertible green-to-red Cac channel (Cac-Maple) was generated using CRISPR by inserting Maple onto the C-terminal region of Cac at the same location previously used for UAS-Cac-GFP (**Kawasaki et al., 2004**) and Cac-GFP[C] (**Figure 3**). Cac-Maple localizes to AZs and is visible at all release sites, displaying normal heterogeneity across the AZ population. Prior to photoconversion, Cac-Maple is entirely green without any red signal. Upon exposure to 405 nm light, the protein completely and irreversibly photoconverts to red (**Figure 7—figure supplement 1A**). TEVC recordings demonstrated that Cac-Maple supports normal

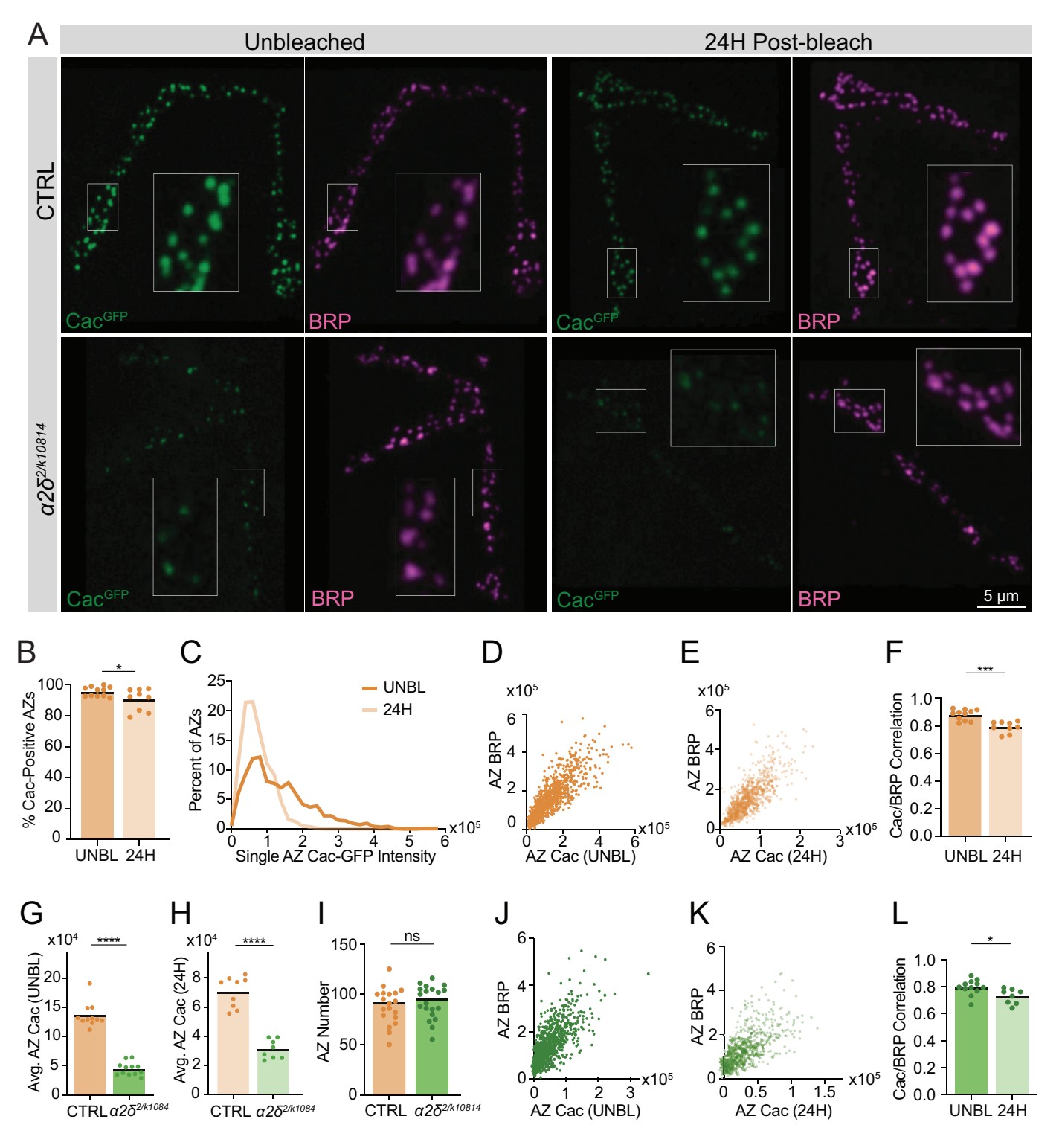

**Figure 6.** Cac delivery to active zones (AZs) correlates with AZ size and is regulated by α2δ. (**A**) Representative images of FRAP experiments in CTRL (*cac^GFP*) and α2δ mutant (*cac^GFP*;*α2δ^2/k10814*) neuromuscular junctions (NMJs). Fixed and Bruchpilot (BRP)-immunostained (magenta) NMJs are shown 24 hr post-bleaching, with unbleached NMJs as a control. Boutons inside white boxes are enlarged. (**B**) Percent of BRP-positive AZs with Cac-GFP max pixel intensity greater than 150% of average background fluorescence (pre: 95.56±0.9347; 0H: 90.37±2.347, p=0.0422). (**C**) Distribution of single AZ Cac-GFP intensity (sum Cac-GFP fluorescence within an ROI encompassing each BRP-positive AZ) in unbleached NMJs (UNBL) and 24 hr after bleaching (pre: 13,6052±3018, n=946 AZs; 24H: 72,261±1389, n=804 AZs). (**D, E**) Correlations of Cac-GFP and BRP at individual AZs in unbleached controls (**D**) and

*Figure 6 continued on next page*

*Figure 6 continued*

NMJs 24 hr post-bleach (**E**). (**F**) Pearson r value for the correlation of Cac-GFP and BRP intensity across the AZ population at each NMJ in unbleached control NMJs (UNBL) and 24 hr post-bleach (pre: 0.8801±0.01191; 24H: 0.7913±0.01348, p<0.001). (**G**) Average AZ Cac-GFP intensity in unbleached segments in $cac^{GFP}$ and $\alpha2\delta^{2/k10814}$ NMJs ($cac^{GFP}$: 136,462±6346, n=11 NMJs; $\alpha2\delta^{2/k10814}$: 44,340±3252, n=12 NMJs, p<0.0001). (**H**) Average AZ Cac-GFP intensity 24 hr post-bleach ($cac^{GFP}$: 70,528±3508, n=9 NMJs; $\alpha2\delta^{2/k10814}$: 31,086±2326, n=8 NMJs, p<0.0001). (**I**) AZ number at NMJs ($cac^{GFP}$: 88.26±4.13, n=19 NMJs; $\alpha2\delta^{2/k10814}$: 92.45±3.689, n=20 NMJs). (**J, K**) Correlation of Cac-GFP and BRP at individual AZs in unbleached $\alpha2\delta^{2/k10814}$ mutants (**J**) and $\alpha2\delta^{2/k10814}$ NMJs 24 hr post-bleach (**K**). (**L**) Pearson r value for the correlation of Cac-GFP and BRP intensity across the AZ population in unbleached NMJs (UNBL) and NMJs 24 hr post-bleach in $\alpha2\delta^{2/k10814}$ mutants (UNBL: 0.7952±0.01635; 24H: 0.7285±0.01955, p=0.018).

The online version of this article includes the following source data and figure supplement(s) for figure 6:

**Source data 1.** Source data for *Figure 6*.

**Figure supplement 1.** Photobleaching validation and GluRIIA correlations.

**Figure supplement 1—source data 1.** Source data for *Figure 6—figure supplement 1*.

evoked synaptic transmission both before and after photoconversion (*Figure 7—figure supplement 1B-E*).

Early 2nd instar $cac^{Maple}$ larvae were photoconverted for a total of 60 s under a mercury lamp with a 405 nm filter and animals were dissected and imaged either 24 hr or 5 days post-conversion. Since Maple is brighter and more photostable in its red state, green BRP immunostaining was used to facilitate reliable AZ identification in experiments measuring red Cac-Maple decay across time. In animals 24 hr post-conversion, a majority of AZs were labeled with red Cac-Maple, with a minority of AZs lacking red signal (*Figure 7A*). This minority represents that AZs formed after the photoconverted pool was AZ-incorporated. In animals imaged 5 days post-conversion, the population of red-labeled AZs was present among a much larger population of green-only AZs, as expected given AZ number doubles daily during this period of NMJ growth (*Akbergenova et al., 2018*; *Figure 7A and B*). Occasionally, entire boutons or long inter-bouton spaces were completely devoid of red channels, representing parts of the NMJ that grew after photoconversion (*Figure 7A*, white arrow).

The red signal present at AZs 5 days post-conversion represents the fraction of remaining Cac-Maple channels that were either present at AZs at the time of conversion, or were photoconverted en route to AZs and incorporated shortly after conversion. To avoid drawing an arbitrary cutoff for 'zero' red signal, the measurement of Cac-Maple decay was performed at the brightest 30 AZs per NMJ (see Materials and methods). The average fluorescence intensity of photoconverted red Cac-Maple at these AZs (per NMJ) decayed by 28–30%, with a predicted Cac $T_{1/2}$ of 8 days at 18°C (*Figure 7C and D*, *Figure 7—figure supplement 1F*). The total number of red-labeled AZs did not change between day 1 and day 5 post-conversion, suggesting that Cac at AZs is stable and does not undergo lateral transfer between neighboring AZs (*Figure 7B*). Moreover, many bright red AZs were observed with neighboring AZs entirely lacking red Cac-Maple signal, further supporting the local stability of Cac at individual AZs (*Figure 7A*).

## Cac turnover at AZs is abolished by reductions in either Cac or α2δ expression

To directly assay whether α2δ plays a role in facilitating Cac surface retention, we measured Cac-Maple turnover over 4 days in an $\alpha2\delta^{2/+}$ heterozygous background ($cac^{Maple};\alpha2\delta^{2/+}$) and in the $\alpha2\delta^{2/k10814}$ transheterozygote ($cac^{Maple};\alpha2\delta^{2/k10814}$). In control NMJs ($cac^{Maple}$), red Cac-Maple intensity diminished by 30% over the 4-day window (*Figure 7A, C and D*). However, measurable Cac turnover at AZs was eliminated in both $\alpha2\delta^{2/+}$ and $\alpha2\delta^{2/k1081}$ backgrounds, as red Cac-Maple intensity was unchanged at AZs on day 5 compared to day 1 (*Figure 7A and D*). The lack of turnover in α2δ reduction backgrounds indicates α2δ is not facilitating Cac abundance at AZs by promoting stabilization. In addition, it suggests that reduced levels of Cac delivery to AZs in these mutant backgrounds decreases Cac turnover. If Cac turnover is dependent on Cac delivery, then animals heterozygous for the Cac deficiency ($cac^{Maple/Df}$), which mildly reduced AZ Cac levels, should have reduced turnover when compared with a $cac^{Maple}$ homozygous control. Unlike Cac in control AZs, which decayed by 30% over a 4-day window, Cac-Maple in the $cac^{Maple/Df}$ heterozygote did not show measurable levels of decay, phenocopying the α2δ reduction mutants and supporting a model in which Cac turnover depends on new Cac delivery to AZs (*Figure 7A and D*). The fact that reducing either α2δ or Cac abolishes Cac turnover demonstrates

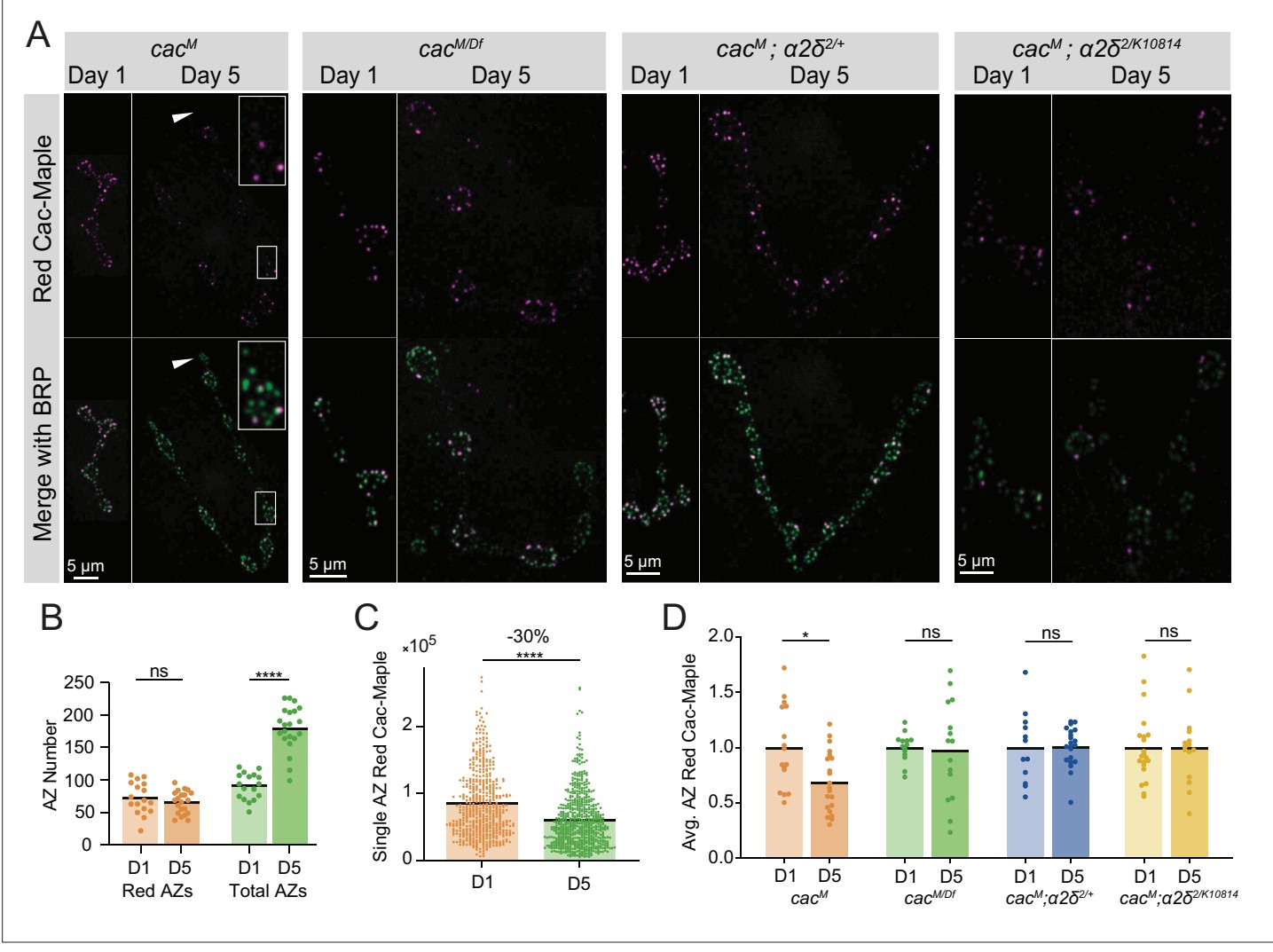

**Figure 7.** Cac turnover at active zones (AZs) is abolished by reductions in either Cac or α2δ expression. (**A**) Representative images of endogenously tagged photoconverted Cac-Maple (magenta) and Bruchpilot (BRP) (green, overlapping with green unconverted Cac-Maple for reliable AZ identification) in controls ($cac^M$), $cac^{M/Df}$ heterozygotes, $α2δ^{2/+}$ heterozygotes ($cac^M;α2δ^{2/+}$), and $α2δ^{2/k10814}$ mutants ($cac^M;α2δ^{2/k10814}$) at 1 and 5 days after photoconversion. Boutons in white box are enlarged in control panel, and white arrows mark a bouton with no red Cac-Maple fluorescence. (**B**) Number of red-marked AZs (photoconverted Cac-Maple) and total AZs (identified using BRP immunostain) quantified 1 and 5 days post-conversion (red D1: 72.59±5.921, n=17 NMJs; red D5: 65.57±3.789, n=21 NMJs; total D1: 90.65±4.948; total D5: 179.5±7.502, p<0.0001). (**C**) Red Cac-Maple fluorescence at individual AZs (30 brightest AZs per NMJ pooled from 16 to 21 NMJs) at 1 and 5 days post-photoconversion. Percent reduction in brightness at day 5 compared to day 1 is shown above (day 1: 84,672±2258, n=502 AZs; day 5: 59,383±1598, n=630 AZs, p<0.0001). (**D**) Average Cac-Maple fluorescence of the 30 brightest AZs per NMJ at indicated genotypes at 1 and 5 days post-conversion. For each genotype, average Cac-Maple intensity at day 1 was normalized to 1.0 ($Cac^M$ D5: 0.6868±0.05975, p=0.0103; $cac^{M/Df}$ D5: 0.9746±0.1154; $Cac^M;α2δ^{2/+}$ D5: 1.006±0.03891; $Cac^M;α2δ^{2/k10814}$ D5: 1±0.08489).

The online version of this article includes the following source data and figure supplement(s) for figure 7:

**Source data 1.** Source data for *Figure 7*.

**Figure supplement 1.** Cac-Maple localizes to active zones (AZs), photoconverts fully, and has normal function.

**Figure supplement 1—source data 1.** Source data for *Figure 7—figure supplement 1*.

that turnover is driven by new delivery and that α2δ is required for Cac delivery. A minor secondary role for α2δ in stabilizing Cac at AZs cannot be completely ruled out due to the residual levels of α2δ remaining in these mutant backgrounds.

In a model where α2δ levels rate-limit Cac delivery to AZs and Cac is normally expressed in excess, Cac overexpression should not result in increased Cac delivery to AZs and Cac turnover should be unaffected by Cac overexpression. UAS-Cac-GFP overexpression using a pan-neuronal driver

(elav-GAL4) in the *cac^Maple* background did not alter Cac-Maple turnover rate despite a reduction in initial Cac-Maple abundance at AZs as expected from competition between the two pools (*Figure 7—figure supplement 1G, H*). These data further indicate that Cac overexpression does not increase Cac delivery to AZs, as suggested by imaging and electrophysiology described above (*Figure 3*). Together, these data indicate that Cac delivery to growing AZs drives Cac turnover, with α2δ promoting Cac delivery into AZs rather than facilitating Cac retention at release sites.

## Discussion

In the present study, we used the *Drosophila* NMJ as a model synapse to characterize the flow of VGCCs into and out of AZs, probing the relationship between Cac, its conserved regulatory subunit α2δ, and the AZ scaffold. Unlike extensive research that has been performed on the trafficking and retention of postsynaptic glutamate receptors at rest and during plasticity, much less is known about how the key engine of $Ca^{2+}$ influx for presynaptic release, VGCCs, are delivered, accumulate, and recycle at AZs (*Rasse et al., 2005*; *Citri and Malenka, 2008*; *Hastings and Man, 2018*). Because of the steep dependence of SV fusion on $Ca^{2+}$ influx and the high correlation between VGCC abundance and $P_r$ at individual AZs, understanding how VGCC abundance is established through delivery and turnover provides critical insights into how neurons regulate synaptic strength (*Akbergenova et al., 2018*; *Gratz et al., 2019*; *Augustine et al., 1985*; *Borst and Sakmann, 1996*; *Sheng et al., 2012*). In the current study, we find that Cac channels are not essential for AZ formation at *Drosophila* NMJs. We also demonstrate that AZs are buffered against moderate increases or decreases in Cac biosynthesis, in contrast to the lack of buffering for the AZ scaffold BRP. We find that Cac delivery to AZs occurs broadly across the AZ population, correlates with AZ size, and is regulated by α2δ. Finally, we show that Cac recycling from AZs is promoted by new Cac delivery and counterbalances delivery at mature AZs to generate the observed upper limit on Cac accumulation at single release sites.

Despite the well-established requirement of VGCCs in synaptic function, whether they also promote AZ structural formation at *Drosophila* synapses has not been determined. Using single-neuron knock-outs, we showed that the α1 subunit of the *Drosophila* VGCC Cac is dispensable for AZ formation and structural maturation. A recent study in mammalian neurons used a similar approach to demonstrate normal AZ number and structure despite conditional knockout of the entire $Ca_v2$ family of VGCCs in cultured mouse hippocampal neurons and the Calyx of Held (*Held et al., 2020*). Many molecular components of the AZ are functionally conserved, but the broad variety of AZ architectures across species raises interesting questions about the similarity of AZ formation and maturation processes (*Zhai and Bellen, 2004*). The finding that AZ formation is independent of VGCCs at *Drosophila* NMJs and mammalian central synapses suggests that this is likely to be a conserved feature of neurons across evolution. While Cac is not essential for BRP accumulation at growing AZs, BRP plays an established role in promoting proper Cac clustering and accumulation (*Fouquet et al., 2009*; *Kittel et al., 2006*). However, whether BRP plays a rate-limiting role in regulating Cac abundance at the AZ is unclear. Here, we demonstrate that Cac AZ abundance is unaffected by a 36% reduction in AZ BRP (BRP heterozygote deficiency background) or by ~80% reduction in AZ BRP in the BRP-positive subset of AZs present following BRP RNAi knockdown (*Figure 4*, *Figure 4—figure supplement 1*). While it is possible that a subset of BRP proteins are post-translationally modified to rate-limit Cac accumulation at AZs, these findings suggest that BRP's presence or absence, rather than its abundance, controls whether AZs accrue normal levels of Cac.

Because the BRP/RBP AZ scaffold forms independent of Cac, we predicted that Cac accumulation at growing AZs is regulated independently from other AZ components. Many synaptic proteins co-traffic to the synaptic terminal, but VGCCs have not been identified on these transport vesicles (*Vukoja et al., 2018*; *Wu et al., 2013*; *Maas et al., 2012*; *Bury and Sabo, 2011*; *Maeder et al., 2014*). We focused on levels of the *Drosophila* CASK/ELKS homolog BRP as a representative AZ scaffold protein given *brp* mutants lack AZ T-bars and show a significant reduction in Cac and other key scaffolding proteins at release sites (*Kittel et al., 2006*; *Fouquet et al., 2009*). In long-lived larvae, we observed increased BRP enrichment at AZs and a leveling-off of Cac accumulation. This divergent trajectory of Cac and BRP accumulation over an extended developmental window suggests that the two proteins are independently regulated.

Manipulation of BRP biosynthesis revealed that BRP AZ abundance directly reflects its biosynthesis, consistent with observations that a mutant with increased AZ seeding (*endophilin*) and one with fewer

mature AZs (*rab3*) have opposite changes in AZ BRP abundance (*Goel et al., 2019*). In contrast to BRP, we found that AZ Cac levels do not reflect Cac biosynthesis. Moderate reductions in Cac biosynthesis do not result in comparable depletion of Cac at AZs, indicating that AZs are buffered against alterations in Cac transcription or translation. Kenyon cells in the *Drosophila* mushroom body maintain normal presynaptic release following weak Cac RNAi knockdown, but fail to potentiate presynaptic release during odor conditioning, suggesting that Cac buffering also occurs in the CNS and its biosynthesis may become rate-limiting during certain forms of presynaptic plasticity (*Stahl et al., 2022*). We also find that Cac overexpression fails to increase Cac abundance at AZs, indicating that Cac biosynthesis does not rate-limit AZ Cac accumulation and suggesting that competition between Cac channels for AZ localization. To visualize this competition directly, we overexpressed red-tagged UAS-Cac-tdTomato in the CRISPR Cac-GFP background and saw that red channels displace green channels from AZs. This competition between channels is supported by VGCC overexpression experiments in mammalian neurons that demonstrate overexpression of PQ channels with channelopathy mutations compete with wildtype channels for AZ localization, and that overexpressing the Ca$_v$2 α1 subunit fails to increase Ca$_v$2 levels at AZs (*Cao et al., 2004*; *Cao and Tsien, 2010*; *Hoppa et al., 2012*).

The two major processes downstream of Cac biosynthesis that establish Cac AZ levels are Cac delivery to AZs and Cac stabilization once incorporated. However, the patterns and rate of Cac delivery to AZs have not been described in detail. We tested whether Cac delivery to AZs occurs in a targeted fashion (with delivery to a subset of AZs) or a population-wide manner (where all AZs within a terminal have continuous access to new channels throughout development). Quantification of Cac delivery at individual AZs revealed a majority of AZs receive new channels over 24 hr, but the levels of new Cac delivery were well below AZ Cac capacity, consistent with previous reports that AZs mature to high-releasing states over a period of several days (*Rasse et al., 2005*; *Akbergenova et al., 2018*). Because AZs level off their accumulation of Cac over developmental time, we expected to find reduced or halted Cac delivery at the most mature AZs. Surprisingly, the largest AZs continued to receive the most Cac, as delivery correlated highly with GluRIIA and BRP levels (two reliable correlates of AZ age and maturation state) across the entire AZ population (*Akbergenova et al., 2018*). We propose a model where new Cac channels are available to the entire AZ population rather than being targeted to a subset of AZs, and larger AZs incorporate more of the available channels than smaller AZs. This distribution of Cac delivery differs from postsynaptic glutamate receptor fields at the *Drosophila* NMJ, where new receptors are only added to growing PSDs, while PSDs that reach a mature state do not incorporate new receptors (*Rasse et al., 2005*).

If mature AZs level off in their Cac accumulation despite increasing levels of Cac delivery, Cac turnover must be occurring at mature AZs. VGCC turnover from the presynaptic membrane has not been measured in any in vivo system, leaving many open questions about both the inter-AZ mobility and stability of Cac channels, as well as their AZ resident time. We generated endogenously tagged green-to-red photoconvertible Cac-Maple to analyze resident time of endogenous Cac channels at AZs and found that roughly 30% of Cac channels are removed from AZs over 4 days at 18°C, suggesting an average half-life of 8 days under these conditions. This turnover was significantly reduced when the transcription of either Cac or the VGCC trafficking subunit α2δ were lowered, indicating that Cac turnover at AZs is promoted by new Cac delivery rather than Cac channels being removed from AZs with a set half-life. The lack of observed turnover in Cac and α2δ reduction backgrounds also indicates that Maple photobleaching is not responsible for the 30% reduction in red Cac-Maple signal measured in wildtype animals. These data support a model where Cac channels turn over at AZs more readily when Cac delivery approaches or exceeds Cac capacity, consistent with the idea that channels compete for spots at AZs, perhaps through limited stabilization by binding partners.

Cac channels could either be removed from AZs through degradation, or undergo lateral transfer and stabilization into nearby AZs. Though we cannot rule out that limited lateral transfer may occur below our limit of detection, two observations suggest lateral transfer is unlikely to be a main avenue of Cac loss from individual AZs. First, the number of red-labeled AZs remained constant over the 4-day window in photoconverted Cac-Maple larvae. If red-labeled Cac channels were drifting into nearby AZs, one would expect an increase in red-labeled AZs. Second, at 4 days post-photoconversion, brightly red-labeled AZs are typically surrounded by AZs with no detectable red signal, suggesting red-labeled Cac does not drift between neighboring AZs during this developmental window. Consistent with this model, new studies using mEOS tagging to track the motion of individual Cac channels revealed a

high degree of Cac mobility within individual AZs, but no significant lateral diffusion between AZs (*Ghelani et al., 2022*). This observation parallels findings from the postsynaptic compartment, where new glutamate receptor fields form via new delivery instead of splitting from existing receptor fields (*Rasse et al., 2005*).

The ability to separately measure Cac delivery and turnover provides new approaches to understand how VGCC regulatory proteins control synaptic abundance of Cac. Here, we focused on α2δ – a conserved VGCC subunit and a target of the commonly prescribed drugs pregabalin and gabapentin (*Dolphin, 2016*; *Dickman et al., 2008*; *Hoppa et al., 2012*; *Eroglu et al., 2009*). α2δ's canonical and highly conserved role is to promote surface expression of the VGCC's pore-forming α1 subunit (*Hoppa et al., 2012*; *Saheki and Bargmann, 2009*; *Cassidy et al., 2014*; *Dickman et al., 2008*; *Ly et al., 2008*). Indeed, we observed that α2δ is required for AZ Cac abundance in a dosage-sensitive manner. While Cac overexpression failed to increase Cac abundance at AZs, α2δ overexpression was sufficient to shift the AZ population toward higher Cac abundance without increasing Cac at the most mature AZs. α2δ is thought to facilitate VGCC surface expression through promoting forward trafficking rather than inhibiting AZ retention. Our data provides new support for this forward trafficking model. First, α2δ overexpression cannot drive increased AZ Cac levels without excess Cac translation, indicating that α2δ is required between translation and AZ delivery, rather than via a post-delivery mechanism. Second, α2δ mutants show reduced Cac delivery but increased Cac stability. A well-described VGCC-independent role of α2δ is to promote synapse formation (*Kurshan et al., 2009*). Our data support a model that α2δ's role in synapse formation is independent of VGCC localization, as neurons form AZs and grow at a normal rate in the absence of VGCCs, but severe reductions in α2δ cause a reduction in synaptic growth and AZ formation.

In summary, these data provide a model of Cac regulation in which Cac abundance at AZs is buffered against moderate alterations in Cac biosynthesis and Cac accumulation is rate-limited by α2δ-mediated trafficking (*Figure 8A*). Cac incorporation into AZs correlates with AZ size, and new delivery promotes Cac turnover at mature AZs, allowing AZ Cac accumulation to level off after several days of development. Our findings indicate that VGCC abundance at AZs reaches a capacity limit, similar to the previously proposed 'slot' model (*Cao et al., 2004*). Future work is needed to identify the molecular basis for this limited capacity and where excess VGCCs reside within the neuron. In addition to AZ-localized Cac, immobile puncta of endogenously tagged Cac-GFP are enriched in axon bundles proximal to the ventral nerve cord (VNC; *Figure 8B*). Following Cac-GFP overexpression, VNC-proximal axonal regions show a roughly 40 µm long area of Cac-GFP enrichment, beginning approximately 25 µm from the soma, likely corresponding to the distal axon segment, an invertebrate parallel to the axon initial segment (AIS) in mammals, defined by enrichment for the para sodium channel (*Ravenscroft et al., 2020*; *Figure 8C*). It is unknown whether this Cac population has a functional role in the distal axon segment, or may instead represent an excess pool of the protein that is available for AZ delivery following mobilization. Clarifying the localization and regulation of nonsynaptic Cac channels may also provide insight into plasticity mechanisms, since the abundance of Cac and other AZ proteins are reported to change during short- and long-term plasticity at *Drosophila* NMJs (*Gaviño et al., 2015*; *Gratz et al., 2019*; *Böhme et al., 2019*; *Hong et al., 2020*). Future studies should reveal additional molecular mechanisms that mediate bulk flow of VGCCs to and from AZs, as well as how these processes can dynamically change during synaptic plasticity.

## Materials and methods
### *Drosophila* stocks

Flies were cultured on standard medium and maintained at 18–25°C. Late 3rd instar larvae were used for imaging and electrophysiological experiments unless otherwise noted. Western blots were performed on adult brain extracts. For experiments performed in homozygous *cac^Maple* and *cac^GFP* backgrounds, males were used to simplify genetic crosses unless otherwise noted. Experiments were performed in a *w^1118* (BCSC #3605) genetic background unless otherwise noted. Fluorescently tagged Cac lines include N-terminally CRISPR-tagged Cac-GFP^N (referred to in text as Cac-GFP; *Gratz et al., 2019*); Cac-GFP^C (C-terminally CRISPR tagged, this study); Cac-Maple (C-terminally CRISPR tagged, this study); UAS-Cac-GFP (*Kawasaki et al., 2004*); and UAS-Cac-tdTomato (provided by Richard Ordway). A Cac deficiency (*cac^Df*; BDSC #9171; *Ryder et al., 2007*) was used to generate Cac heterozygotes.

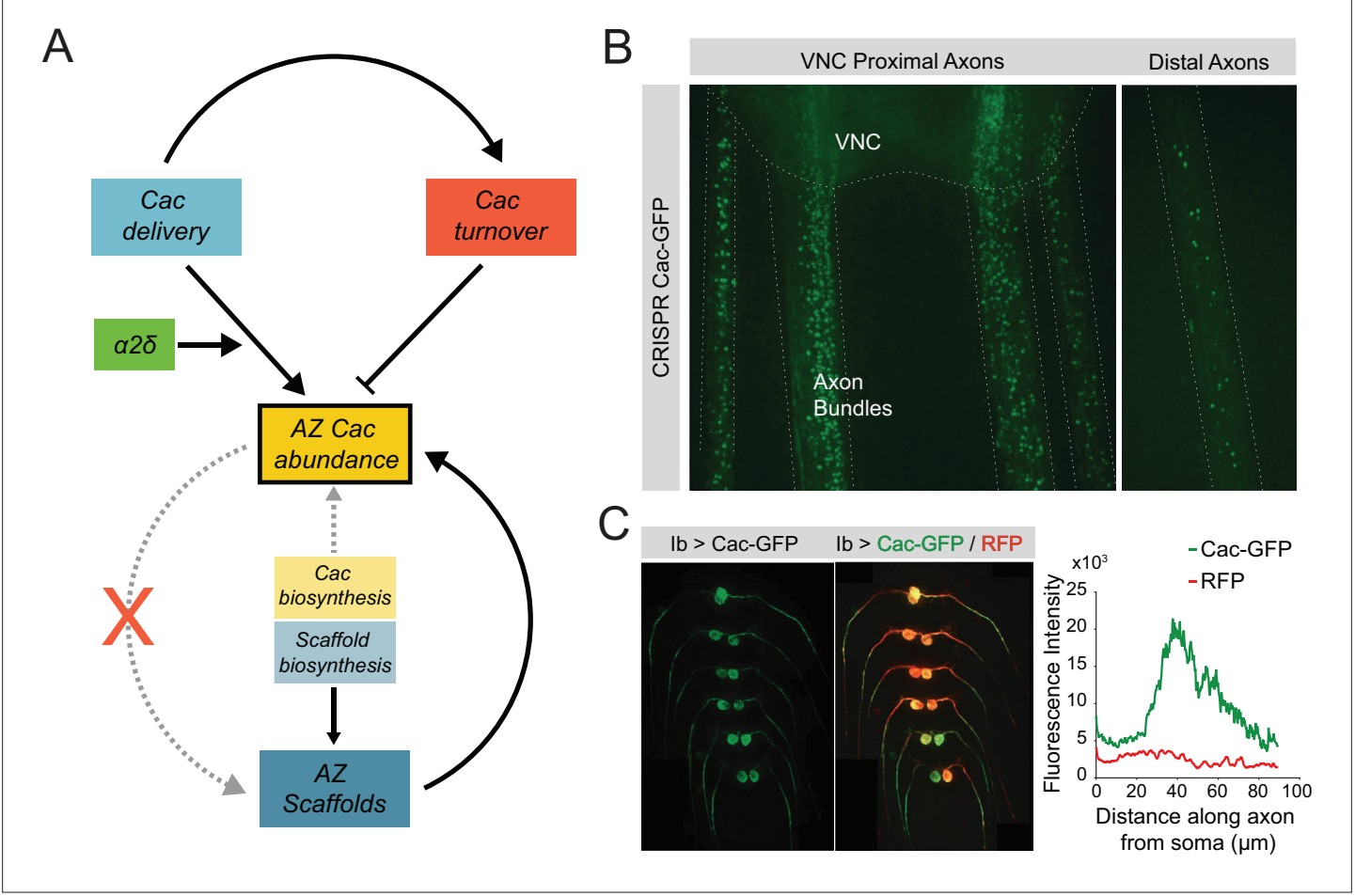

**Figure 8.** Model of Cac and active zone (AZ) scaffold regulation at the *Drosophila* neuromuscular junction (NMJ). (**A**) AZ Cac abundance (yellow box) is regulated by both new Cac delivery (blue box) and Cac turnover from AZs (red box). Cac delivery is positively regulated by α2δ, and new delivery promotes turnover of existing channels. Cac biosynthesis (light yellow box) weakly regulates AZ Cac abundance, as AZ abundance is buffered against moderate changes in biosynthesis. In contrast, scaffold biosynthesis is a strong regulator of AZ scaffold abundance. The dependence relationship between AZ Cac abundance and AZ scaffold abundance is unidirectional: AZ scaffolds regulate Cac accumulation, but Cac is dispensable for AZ scaffold formation. (**B**) Left – representative image of endogenously tagged Cac-GFP puncta in axon bundles proximal to the ventral nerve cord (VNC). Right – representative image of Cac-GFP in distal axons. These axon puncta are immobile over a period of 30 min. (**C**) Left – representative image of a VNC with Cac-GFP (green) and RFP (red) expressed exclusively in MN1-Ib neurons. Right – quantification of Cac-GFP and RFP signal intensity as a function of distance along the axon from the soma. Plotted signal intensity was averaged across eight axons. RFP intensity is constant throughout the first 100 μm, while Cac-GFP intensity is elevated in the 30–60 μm range.

For single-neuron Cac knockout experiments, the Ib-specific Gal4 driver GMR94G06 (BDSC #40701) was used to drive UAS-FLP (BDSC #4539) in the $cac^{flp.nd}$ background (BDSC #67681; *Fisher et al., 2017*). Long-lived larval experiments were performed using one copy of phm-Gal4 and one copy of UAS-torso-RNAi (*Rewitz et al., 2009*; *Miller et al., 2012*). BRP mutants and RNAi constructs used for this study include $brp^{69}$ (provided by Stephan Sigrist), $brp^{Df}$, and BRP RNAi (provided by Ethan Graf). α2δ (straightjacket) mutants and constructs used for this study include $stj^2$ (BDSC #39715); $stj^{k10814}$ (BDSC #11004), and UAS-Stj (FlyORF F001495). PSD labeling for intravital FRAP experiments was performed using GluRIIA-RFP inserted onto chromosome III under the control of its endogenous promoter (provided by Stephan Sigrist). Pan-neuronal expression was performed using $elav^{C155}$-GAL4 (BDSC #8765).

## Generation of CRISPR-tagged Ca²⁺ channel lines

Two endogenously C terminally tagged Cac lines (Cac-GFP$^C$ and Cac-Maple) were generated in this study using the same CRISPR genome engineering approach. One guide RNA (

CGAGGGTTCAGACCACTCTT) was chosen using the CRISPR Optimal Target Finder (*Gratz et al., 2014*) and inserted twice into the pCFD4 expression vector (Addgene #49411, *Port et al., 2014*) using the Gibson assembly protocol and NEBuilder HighFidelityDNA Assembly Cloning Kit (E5520). Donor template constructs were generating using Gibson assembly. Donor templates encode one Kb 5' and 3' homology arms flanking the 3' end of the *cac* coding region; these arms were generated through PCR amplification from the *Drosophila* genome. The 5' homology arm was mutated at the gRNA-binding site using silent mutations that do not alter the amino acid sequence (mutated to CGAGGGTTCAGACCtgatTT). DNA encoding the fluorescent tag (either GFP or Maple) was inserted between the homology arms and flanked by the restriction sites EcoRI and XbaI. The tag was inserted immediately before the stop codon, with the same tagging location and GFP identity previously used to generate UAS-Cac-GFP (*Kawasaki et al., 2004*). A portion of the pCFD4 vector was linearized using PCR (forward primer: caattgtgctcggcaacagt; reverse primer: caattgatcggctaaatggtatg) and used as a backbone sequence for the donor template plasmid. gRNA and template plasmids were co-injected into *yw;;nos-Cas9*(III-attP2) embryos by BestGene Inc (Chino Hills, CA). PCR was used to screen progeny of injected animals for presence of the fluorescent tag. The modified locus was confirmed by sequencing.

## Immunocytochemistry

Larvae were dissected in HL3 solution and fixed in 4% paraformaldehyde for 10 min, washed in $Ca^{2+}$-free HL3, and blocked in 5% normal goat serum and 5% BSA in PBT for 15 min. Samples were incubated overnight at 25°C in blocking solution containing primary antibodies, and then washed for 1 hr in blocking solution. Samples were incubated for 1–2 hr at room temperature in blocking solution containing fluorophore-conjugated secondary antibodies. Primary antibodies used in this study were mouse anti-BRP at 1:500 (Nc82 DSHB, Iowa City, IA), rabbit anti-RBP at 1:500 (provided by Stephan Sigrist), mouse anti-SYX1 at 1:100 (8C3; DSHB), rabbit anti-GluRIII at 1:500 (*Marrus et al., 2004*), and anti-CPX at 1:500 (*Huntwork and Littleton, 2007*). Secondary antibodies used in this study were goat anti-mouse Alexa Fluor 607-, 546-, or 488-conjugated anti-mouse IgG at 1:5000 (Invitrogen, #s A21237, A11030, and A32723) and goat anti-rabbit Alexa Fluor 488-conjugated IgG at 1:5000 (A-11008; Thermofisher). For HRP staining, samples were incubated in DyLight 649-conjugated HRP at 1:500 (#123-605-021; Jackson Immuno Research, West Grove, PA). Samples were mounted in Vectashield (Vector Laboratories, Burlingame, CA). For Cac Flpstop experiments, red tdTomato reporter fluorescence was completely abolished by fixation in 4% paraformaldehyde, allowing other proteins to be immunostained and imaged post-fixation in the red channel.

## Cac-Maple photoconversion and analysis

To photoconvert larvae, early 2nd instar animals were placed on a glass slide in a thin film of halocarbon oil to prevent drying. Animals freely crawled for three 20 s intervals under a mercury lamp with a 405 nm filter. After photoconversion, animals were placed in chambers with food in an 18°C incubator and were kept in the dark to prevent photobleaching from ambient light. Animals were dissected and stained with anti-BRP with Alexa Fluor 488-conjugated secondary antibody, and imaged under oil immersion. The sum fluorescence of red Cac-Maple signal within ROIs drawn manually to encompass the entire red-positive AZ area was determined. For each AZ ROI, mean single-pixel background fluorescence was multiplied by ROI pixel number and subtracted. The 30 AZs per NMJ with the brightest red Cac-Maple signal were analyzed to avoid AZs near the lower brightness limit of detection. Analysis on control animals selecting the top 10, 20, or 30 brightest AZs demonstrates the number of brightest AZs analyzed did not affect turnover measurements (*Figure 7—figure supplement 1F*).

## Confocal imaging and imaging data analysis

Confocal images were acquired using a Zeiss Axio Imager 2 with a spinning-disk confocal head (CSU-X1; Yokagawa) and ImageM X2 EM-CCD camera (Hammamatsu). An Olympus LUMFL N 60× objective with a 1.10 NA was used for live imaging, and a Zeiss pan-APOCHROMAT 63× objective with 1.40 NA was used for imaging stained or intravitally mounted animals. A 3D image stack was acquired for each NMJ imaged. Image analysis was performed in Volocity 3D Image Analysis software (PerkinElmer) and 3D stack images were merged into a single plane for 2D analysis. Analysis of Cac, BRP, RBP, and GluRIIA intensities were performed by first identifying AZ position using the 'find spot' algorithm in

Volocity 3.2 software that detects fluorescent peaks, and AZs were manually added in cases where the algorithm missed an AZ. ROIs were automatically generated by the software from identified spots and max pixel intensity or sum pixel intensity within the AZ ROI was reported as noted in figure legends. Background fluorescence was measured using the mean pixel intensity of background (non-NMJ) areas within merged images and background was subtracted from AZ fluorescent measurements. HRP area measurements were performed using the 'find object' algorithm in Volocity 3.2 software that creates an ROI around the HRP-positive area.

## Live intravital imaging and photobleaching

Larvae were anesthetized with SUPRANE (desflurane, USP) from Amerinet Choice (*Zhang et al., 2010*). Larvae were placed on a glass coverslide in a thin film of halocarbon oil to prevent drying and incubated under a glass jar with a small cotton ball soaked in Suprane for 40 s in a fume hood. For imaging, anesthetized larvae were covered in halocarbon oil and positioned ventral side up on a glass slide between spacers made by clay, compressed moderately, and covered with a cover glass. NMJs on muscle 26 in hemi-segment A2 or A3 were imaged. For photobleaching experiments, NMJs were imaged continuously with 80% laser power and 1 s exposure for 90 s until green fluorescent signal was no longer visible. After an imaging session, larvae were placed in individual Eppendorf tubes with food in a 25°C incubator and were wrapped in aluminum foil to prevent photobleaching by ambient light in the room. Animals that did not recover within 5 min of unmounting were excluded from further analysis. The same data acquisition settings were used to visualize NMJs at different larval stages. To facilitate easier visual comparison between developmental time periods, images of the corresponding NMJ area at younger stages were cut, rotated when necessary, and placed onto a black background for figure presentation.

## TEVC electrophysiology

Postsynaptic currents were recorded using TEVC with a –80 mV holding potential. Experiments were performed in room temperature HL3.1 saline solution (in mM, 70 NaCl, 5 KCl, 10 NaHCO$_3$, 4 MgCl$_2$, 5 trehalose, 115 sucrose, 5 HEPES, pH 7.18) as previously described (*Jorquera et al., 2012*). Final [Ca$^{2+}$] was adjusted to 0.3 mM unless otherwise noted. Recordings were performed in 3rd instar larvae at muscle 1 of segments A3 and A4 (for Cac knockout experiments) and at muscle 6 of segment A4 (for all other experiments). Motor axon bundles were cut and suctioned into a glass electrode and action potentials were stimulated at 0.33 Hz (unless indicated) using a programmable stimulator (Master8, AMPI; Jerusalem, Israel). Data acquisition and analysis was performed using Axoscope 9.0 and Clampfit 9.0 software (Molecular Devices, Sunnyvale, CA) and inward currents are labeled on a reverse axis.

## Western blot analysis

Adult head lysates were prepared by freezing whole flies in liquid nitrogen in Eppendorf tubes and vortexing briefly to mechanically separate heads from bodies. Heads were sorted on a cold glass slide and five heads per sample were placed into loading buffer (2× Laemmli buffer with 1 M DTT) and homogenized. After centrifugation, supernatant was diluted into water for a final concentration of 1× Laemmli buffer and one head per 10 µl of sample. Western blotting was performed using standard laboratory procedures with mouse anti-SYX1 at 1:1000 (8C3; DSHB), mouse anti-BRP at 1:5000 (NC82; DSHB), rabbit anti-GFP at 1:5000 (ab6556; Abcam, Cambridge, UK) and mouse anti-Tubulin at 1:1,000,000 (Sigma: T5168). IRDye 680LT-conjugated goat anti-mouse at 1:5000 (LICOR; 926-68020) and IRDye 800CW-conjugated goat anti-rabbit at 1:5000 were used as secondary antibodies. A LI-COR Odyssey Imaging System (LI-COR Biosciences, Lincoln, MA) was used for visualization and analysis was performed using FIJI image analysis software (*Schindelin et al., 2012*). Lanes with damaged spots on the membrane that prevented reliable quantification were excluded from analysis.

## Statistical analysis

Statistical analysis and plot generation were performed using GraphPad Prism (San Diego, CA). The plot function in MATLAB R2020A (MathWorks, Natick, MA) was used to generate average electrophysiological traces. For comparisons of three or more groups of data, one-way ANOVA followed by Šídák's or Dunnett's multiple comparisons test was used to determine significance. For comparisons

of two groups, Student's t-test was used. The mean of each distribution is plotted in figures with individual datapoints also shown. Figure legends report mean ± SEM, and n. Asterisks indicate the following p-values: *, $p \leq 0.05$; **, $p \leq 0.01$; ***, $p \leq 0.001$; ****, $p \leq 0.0001$. Pearson coefficient of correlation was calculated in GraphPad Prism using the following parameters: two-tailed p value and 95% confidence interval.

## Acknowledgements

This work was supported by NIH grants MH104536 and NS117588 to JTL. KLC was supported in part by NIH pre-doctoral training grant T32GM007287. We thank the Bloomington *Drosophila* Stock Center (NIH P40OD018537), the Developmental Studies Hybridoma Bank, Ethan Graf (Amherst University), Gerald Rubin (Janelia Research Campus), Kate O'Connor-Giles (Brown University), Richard Ordway (Penn State University), Barry Ganetzky (University of Wisconsin), and Stephan Sigrist (Freie Univesitat Berlin) for providing *Drosophila* strains and antibodies, Dina Volfson for cloning assistance, Ellen Hill for help with Western data acquisition, and members of the Littleton lab for helpful discussions and comments on the manuscript.

## Additional information

### Funding

| Funder | Grant reference number | Author |
|---|---|---|
| National Institute of Mental Health | MH104536 | J Troy Littleton |
| National Institute of Neurological Disorders and Stroke | NS117588 | J Troy Littleton |
| National Institutes of Health | T32GM007287 | Karen L Cunningham |

The funders had no role in study design, data collection and interpretation, or the decision to submit the work for publication.

### Author contributions

Karen L Cunningham, Conceptualization, Data curation, Formal analysis, Investigation, Methodology, Writing - original draft, Writing - review and editing; Chad W Sauvola, Data curation, Formal analysis; Sara Tavana, Data curation; J Troy Littleton, Conceptualization, Supervision, Funding acquisition, Writing - original draft, Project administration, Writing - review and editing

### Author ORCIDs

Karen L Cunningham (iD) http://orcid.org/0000-0002-7738-0318
J Troy Littleton (iD) http://orcid.org/0000-0001-5576-2887

### Decision letter and Author response

Decision letter https://doi.org/10.7554/eLife.78648.sa1
Author response https://doi.org/10.7554/eLife.78648.sa2

## Additional files

### Supplementary files

• Transparent reporting form

### Data availability

All data generated or analyzed during this study are included in the manuscript and supporting file; Source Data files have been provided for Figures 1 to 7.

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
