## [Editor Report]

This paper will be of interest to a broad range of neurophysiologists as it provides insights into the regulation of presynaptic voltage-gated calcium channel abundance which largely determines presynaptic strength. The findings demonstrate that while VGCC biosynthesis does not play a major role in regulating VGCC abundance at AZs, both trafficking and recycling at active zones are important regulatory steps.

---

## [Decision Letter]

**Decision letter after peer review:**

Thank you for submitting your article "Regulation of presynaptic ca^2+^ channel abundance at active zones through a balance of delivery and turnover" for consideration by *eLife*. Your article has been reviewed by 3 peer reviewers, and the evaluation has been overseen by a Reviewing Editor and Claude Desplan as the Senior Editor. The following individuals involved in the review of your submission have agreed to reveal their identity: Heather Broihier (Reviewer #2); Dion K Dickman (Reviewer #3).

The reviewers have discussed their reviews with one another and all agree that this paper is a strong candidate for publication after revision. The Reviewing Editor has drafted this to help you prepare a revised submission.

Essential revisions:

1) There is some shared concern regarding the interpretation of the data in Figure 1 that cac is not required for AZ assembly, since there may be residual Cac in the experiments. The authors do provide data about how much residual Cac remains after their FLP manipulation, and it is likely that substantial Cac persists in larval motor neurons, so the claim about AZ structure in the absence of Cac should be tested more and probably toned down. There is an antibody that can be used to assess the question of how much Cac remains, and/or presynaptic ca^2+^ imaging as the authors have done previously would be sufficient, in addition to modifying their claims, to address this concern.

2) There are a number of issues where the reviewers felt that further experimental evidence would significantly strengthen the manuscript. Addressing these concerns is important and the authors are encouraged, but not required, to add further experimental data regarding the following points:

– It would be nice to have some more direct readout of (participating) calcium channels (e.g. calcium imaging);

– It would help to see a Western Blot analysis for the Cac overexpression;

– In flies, the consequences of a2d OE haven't been reported (to the best of our knowledge). Hence, it would be great to know whether a2d OE is sufficient to seed synapses and also whether a2d OE increases Cac delivery/recycling to synapses;

– It would help to see more of a functional characterization upon expression of the N-terminally GFP tagged Cac, ideally together with a control and the C-terminally tagged construct as Dion suggests.

3) Please find further suggestions in the three individual reviews below.

*Reviewer #1 (Recommendations for the authors):*

1. A concern relates to the interpretation of the experiment depicted in Figure 1. To circumvent the early embryonic lethality of Cac null mutation, the authors apply a single-cell manipulation of gene expression using a Flpstop knockout approach to abolish gene expression using an MN1-1b specific Gal4 line. While the number of BRP-positive AZs from terminals that had undergone a Flp event revealed no change in single AZ number or BRP levels, it is not clear that this observation is sufficient to conclude that "ca^2+^ channels are not required for AZ scaffold formation or synapse growth". A concern is the temporal expression profile of the Gal4 and how early in development Cac gene expression was stopped. Even without further expression, the high stability of previously expressed Cac can be problematic and the remaining AP-evoked activity indicates that functional channels (although reduced in number) are still present at the time of the experiment. Indeed, the authors show remarkable stability of Cac which is even enhanced under the conditions of reduced Cac expression as would be the case here (no detectable decrease within 5 days in heterozygous condition, Figure 7D). Thus, it can neither be excluded that the AZs present at this time required Cac when they were formed nor that the remaining Cac channels are sufficient to mediate AZ formation. Independent of this ambiguity, the presented data are interesting, but I suggest loosening the conclusion and discussing these circumstances.

2. The authors conclude on independent regulation of Cac from BRP because manipulations of BRP levels do not affect Cac levels to the same extent. In this context, it would be much more informative to directly present the relation between both protein levels at individual AZs. The authors have previously shown a strong correlation between BRP and Cac AZ levels (Akbergenova et al., 2018) and it would be nice to see whether this correlation persists with a different relation (slope) under these conditions or whether the relation is lost for all or a subset of AZs. Indeed, the wealth of data calls for a more sophisticated analysis of the single AZ level and the asymmetric distribution of the single AZ intensities indicates that mean intensity values might not be a good proxy for comparisons. Are the AZs found in the "high-intensity tail" of the fluorescence distribution for both proteins the same? Besides the total abundance of scaffolding proteins assessed by NMJ immunostaining relevant interactions with Cac could depend on a post-translational modification which would make both the levels and modification state relevant for Cac accumulation. This should be discussed.

3. As this is a first report of a newly developed Cac-GFPC knockin fly line the authors should provide a complete basal characterization of synaptic structure/composition and electrophysiological responses compared to the Cac-GFPN and control lines.

4. Bands of TUB and SYX1 of gel in Figure 4c are cut. Please correct and provide further details on the tissue processing for Western blot analysis.

5. An interesting observation the authors should follow up is that it appears that not all AZs are equally affected by reducing alpha2delta levels. Again, a more detailed analysis considering the heterogeneity of AZs would be informative. From the examples shown it appears that particularly the BRP-rich AZs contain higher (near normal) Cac levels. Is that so? The other interesting aspect is that sizable transmission remains even when NMJ immunostainings indicate a reduction of 64%. Considering the high cooperativity between ca^2+^ influx and neurotransmitter release, this is truly unexpected (assuming a dependence on the 4th power would predict a reduction by 99%). Similarly, over-expression increases the NMJ levels of Cac-GFP but not the evoked responses (the effect depicted on the 5th stimulus in Figure 5I/J is rather modest) and unexpectedly the short-term plasticity is unaffected. This could indicate that not all stained Cac-GFP proteins/voltage-gated ca^2+^ channels contribute to ca^2+^ influx. The manuscript could be further strengthened by more directly reading out ca^2+^ signals (e.g. using genetically encoded optical reporters or infused synthetic dyes) in these conditions. The authors should then relate the relative ca^2+^ influx to the Cac-GFP levels of the same NMJ to test whether indeed signals primarily depend on the NMJ levels of Cac or whether participation in ca^2+^ influx can be differentially regulated by another layer.

Demonstration of a mechanism under these conditions would increase the impact of this study.

6. Because Cac recovers with a similar (or even slightly faster) time constant in the FRAP experiments, the authors provide evidence that alpha2delta is unlikely involved in the final step of AZ integration which should be stated more clearly. The overall lower AZ levels and reduced flux into AZs would rather be consistent with a reduced supply pool in this condition. Do the authors find any evidence of this from the experiments depicted in Figure 8 where they show an axonal Cac-GFP pool which may be targeted to the NMJ? If axonal delivery played a role, effects may be augmented for more distant body segments, did the authors test anything along those lines?

*Reviewer #2 (Recommendations for the authors):*

1. In Figure 1, can the authors comment on when Tomato expression is first observed? This would help pinpoint how long the MN was Cac-negative. I have some (exceedingly minor) concern that the lack of a structural phenotype in these mutant MNs might be explained by a very recent loss of Cac.

2. Also, regarding Figure 1, their data indicate Cac is dispensable for AZ formation. But is it possible it could be required for apposition? Do they see any defects in Brp/GluR alignment in this background?

3. In Figure 3, the authors argue that synaptic Cac is buffered against changes in gene dosage. They show by Western blot that loss of one copy of Cac results in roughly 50% Cac protein. They do experiments with Cac OE, but do not demonstrate that these MNs actually express more Cac protein by Western blot as they did for the heterozygotes.

4. In Figure 5, they demonstrate that a2d overexpression results in more Cac per AZ. Is a2d sufficient to nucleate synapses? (Do the authors see more synapses per NMJ).

5. Pertaining to Figures 6 and 7, I think it would strengthen the authors' argument that a2d is rate-limiting for Cac removal if they could show increased delivery/recycling when overexpressing a2d.

*Reviewer #3 (Recommendations for the authors):*

1. Cac requirement in AZ assembly (Figure 1): The authors conclude that "Cac is not abundantly present at Cac-flipped AZs", and further indicate that Cac is necessary to enable proper AZ assembly. However, this interpretation appears to rely solely on electrophysiological recordings? Significant transmission persists in the Cac-flipped motor neurons, and there appears to be no effort to image or quantify how much Cac remains after flipping. Hence, there is no way to know how much Cac perdurance remains in motor neurons after flipping simply by recording EJCs.

Ideally, the Cac-flipping would be done with endogenous tagged Cac alleles, but absent that approach there is an anti-Cac antibody published by David Morton and available the authors might use to assess how much Cac remains at AZs after flipping. In addition (or alternatively), the authors could do presynaptic ca^2+^ imaging in Cac-flipped motor neuron terminals to assess how much Cac-mediated ca^2+^ influx remains. Finally, the authors could look at EM to determine whether AZ ultrastructure is altered by loss of Cac.

It is important to note that this is not a point of real controversy, as the Held et al. 2020 paper did such a rigorous job of showing that Cav2 channels are not important for AZ or synapse assembly, and there is no evidence in *Drosophila*, from this study or from other studies, to dispute this key finding. This is also not a major focus of the current manuscript, so as there doesn't seem to yet be an effective way to eliminate all Cac channels in motor neuron in *Drosophila*, the conclusions in this section about whether Cac is needed for AZ formation should be tempered as it cannot be clearly shown.

2. Cac splice isoforms (Figure 3): The data shown in Figure 3 is quite interesting, and is the first to show to what extent Cac overexpression outcompetes endogenous Cac (labeled by endogenous tags). Can the authors quantify how much endogenously tagged Cac is removed and replaced by overexpressed Cac? Even better would be to quantify transcriptional levels of endogenous vs overexpressed Cac (by qPCR) and correlate this with relative protein levels at AZs. This would be a nice demonstration that transcriptional levels of Cac expression ultimately determine AZ protein levels.

More importantly, it seems the authors can make some rather interesting further conclusions about the role of potential splice isoforms of Cac from these experiments. There are many predicted Cac isoforms encoded by the Cac locus (something like 11?), which the endogenously tagged N-term Cac should label, while the UAS-Cac expresses a single isoform.

Some questions:

A) Can the authors clarify whether their endogenous C-terminal Cac, presumably like the endogenous N-term Cac, is predicted to tag every predicted isoform? In the case it does not, then the authors need to be careful about interpretations in this section as it is possible that not every Cac will be tagged by the C-term tag.

B) Were any functional (electrophysiological) differences found between wild type (all Cac isoforms and/or Cac N-term tag) vs Cac-C-term vs UAS-Cac (single isoform)? This would allow the authors to make a point about to what extent different Cac splice isoforms might contribute significant differences to Pr/diversity, at least as assessed through electrophysiological recordings.

C) Similarly, were different splice isoforms apparent on the anti-GFP blot? It looks like at least 2-3 different bands can be seen that appear to be Cac? Could the C-term Cac blot be compared to determine if there is evidence for major differences in Cac isoform expression?

D) Previous studies have overexpressed Cac and saw no major changes in synaptic strength, already suggesting that Cac abundance at AZs is limited through post-transcriptional mechanisms. This point should be mentioned and the appropriate papers cited. The authors also use the vague term "biosynthesis", rather than a more clear terminology of "transcription" vs "translation" vs "post-translational mechanisms". If the authors use biosynthesis to mean the combined product of transcription, translation, and post-translational regulation, the authors should clearly define this in the manuscript. It is clear from the overexpression data that Cac abundance at AZs is regulated by transcription itself, at least in terms of competition between distinct transcripts, without needing to invoke additional translational/post-translational control.

3. Other pathways/targets that may regulate Cac abundance/dynamics at AZs: The first half of this manuscript nicely confirms and extends much of what was known or expected about the importance of Cac in AZ assembly, and how BRP and α-2 δ relate to Cac. There is a lot of important data in this manuscript, and a reasonable point could be made that additional studies might be outside the scope of the current manuscript. But I would suggest the authors consider the following experiments to extend the importance and impact of this study:

A. Can the authors test the role of the ca^2+^ channel Β subunit in Cac regulation? A simple RNAi might be sufficient, but in contrast to α-2 δ, where a lot was known about its importance in regulating Cac levels at AZs, much less is known about the β subunit (not only in *Drosophila* but in mammalian systems as well).

B. Does Cac delivery or turnover change in conditions of hyper- or hypo-activity? There are numerous ways to manipulate activity in *Drosophila* neurons, and to what extent Cac abundance and turnover are sensitive to activity would add an intriguing dimension to this study.

C. Are there any ideas or possible pathways of how a surplus pool of Cac is then limited and selectively trafficked to be incorporated at AZs? This bottleneck appears to be the key node of regulation, and while α-2 δ seems to be involved, some insight into the motors or transport vesicles would be of major interest.

---

## [Author Response]

Essential revisions:We appreciate the reviewers’ positive comments on our manuscript and their suggestions for improvement. All three reviewers were excited about the study, especially regarding the impact of the findings given the central role of voltage-gated ca^2+^ channels in synaptic transmission. Each suggested several avenues for improving the presentation and extending the science. We address their suggestions in a point-by-point response below.

Suggestions shared by reviewers:

Validation of Cac removal in Flpstop experiments

We appreciate the reviewers’ suggestion to further examine Cac at AZs of Cac-flipped NMJs. Given the strong reduction in EJC amplitude in Flpstop animals, together with doubling of AZ number during each day and no detectable lateral movement of Cac across AZs (measured by Cac-Maple and shown later in the manuscript), we expected AZs formed after the flip event to lack Cac, even though earlier-formed AZs would have some Cac that was made before the Flp event occurred. To characterize this in more detail, the timecourse of Cac-flipping during development was assayed by measuring the percentage of MN1-Ib somas that were positive for the tdTomato reporter in first, second and third instar animals. We observed most motoneurons underwent flip events in or before the first instar stage, with a further slight increase in second instars. We also quantified AZ number at the second instar stage when an average of 91% (~13/14) of MN1-Ib neurons had flipped, and found a five-fold lower AZ number than that measured in third instars. These data indicate 78% of AZs observed in the third instar stage are formed after the Flpstop Cac deletion and are unlikely to contain Cac. We added this new data as Figure 1 – Supplemental figure 1A-D. We also attempted to use the published Cac antibody, as suggested, to quantify Cac levels in Cac-flipped NMJs. While Cac staining (red) at BRP-positive AZs (green) was observed at control NMJs (3 NMJs are shown in Author response image 1), the staining was highly variable and overall unreliable for quantification. A much larger percent of AZs lacked Cac staining in controls compared with what we observe with endogenously tagged Cac-GFP, indicating incomplete labeling by the antisera. In addition, the anti-Cac staining had high background that was often punctate, making identification of Cac-negative AZs difficult. As such, this tool was not a reliable option for us to quantify Cac by immunostaining.

**Author response image 1. sa2fig1:** 

However, to extend our analysis we focused on examining BRP and RBP intensity in newly formed AZs after the Flp event that are likely to lack Cac. We performed a more detailed analysis of the BRP and RBP intensity data across the entire AZ population in control versus Cac-flipped NMJs (newly added data as Figure 1 – Supplemental figure 1E-H). While we originally reported average BRP and RBP AZ intensity for each NMJ, we now include two new analyses. First, we added frequency distributions of BRP and RBP over the entire AZ population (Figure 1 -Supplemental figure 1 E,G). Second, we binned the analysis to separately measure BRP and RBP in newer versus older AZs by analyzing the average BRP and RBP intensity in the bottom versus top 50% of the AZ population (Figure 1 Supplemental figure 1 F,H). AZs in the lower half of the distribution were formed most recently, likely multiple days after the Cac flip event occurred. The finding that this bottom 50% of the AZ distribution is unchanged in BRP abundance with only a slight increase in RBP abundance further supports the model that Cac is not required at an AZ for the AZ scaffold to form. As the reviewers noted, some activity remains in this manipulation due to Cac expressed prior to the Flp event. As such, the role of evoked activity within the entire NMJ for AZ formation and maturation is not addressed by these data. These new experiments and analyses have been added as a supplement to figure 1, and the text has been updated to focus on single-AZ rather than whole-NMJ conclusions.

Characterization and validation of endogenously tagged Cac-GFP-C-terminal tagged line

We did not include additional characterization of this line as the same tagging location and tag identity was electrophysiologically characterized in identically tagged UAS-Cac-GFP rescue experiments (Kawasaki et al., 2004). We also provide an electrophysiological characterization of the endogenous Maple C-terminal-tagged Cac line generated in the current study, where Maple was placed in the exact spot as the C-terminal GFP line we generated (Figure 7 – supplemental figure A-E). Given a reviewer also asked about synapse morphology and AZ number in the endogenous Cac-GFP line, we performed this analysis and included the results in Figure 3 – Supplemental figure 1, showing NMJ growth was not affected and AZ number was very mildly reduced. We note the C-terminal tagged GFP line is only used in one dataset (Figure 3A-C), and all other datasets with endogenously tagged Cac use the N-terminally tagged line from Gratz et al., 2019 or the C-terminal Maple version we generated.

Responses to individual reviewers:

Reviewer 1:

Reviewer 1 described the current findings as “novel, unexpected, and important” and the reviewer requested several clarifications we addressed below.

Points not addressed above:

2. “The authors conclude on independent regulation of Cac from BRP because manipulations of BRP levels do not affect Cac levels to the same extent. In this context, it would be much more informative to directly present the relation between both protein levels at individual AZs. The authors have previously shown a strong correlation between BRP and Cac AZ levels (Akbergenova et al., 2018) and it would be nice to see whether this correlation persists with a different relation (slope) under these conditions or whether the relation is lost for all or a subset of AZs. Indeed, the wealth of data calls for a more sophisticated analysis of the single AZ level and the asymmetric distribution of the single AZ intensities indicates that mean intensity values might not be a good proxy for comparisons. Are the AZs found in the "high-intensity tail" of the fluorescence distribution for both proteins the same?”

Our conclusion that Cac and BRP are independently regulated draws on two sets of data. First, the demonstration that Cac and BRP accumulation show divergent trajectories over extended developmental time indicates Cac levels off at AZs while BRP continues to accumulate beyond control levels. Second, Cac levels are buffered at AZs while BRP levels are not. Our finding that reducing BRP levels by 35% has no effect on Cac levels supports a different part of the model – that BRP enrichment is not a rate-limiting factor in Cac AZ accumulation. AZ Cac and BRP both accrue over developmental time, so their levels correlate at AZs – as the reviewer noted, this has been shown extensively in published work from our lab and others (Akbergenova et al., 2018). Here we argue that this correlation does not imply that BRP levels regulate Cac levels or vice versa. Instead, BRP and Cac can accumulate largely independent of each other’s presence at the AZ, resulting in the observed correlation at the AZ level.

To further explore the relationship between Cac and BRP levels at AZs when BRP is dramatically reduced, as suggested by the reviewer, we performed and added a new dataset with extensive individual AZ analysis of this relationship (included as new Figure 4 – Supplemental Figure 1). This allowed us to extend our analysis of Cac abundance in BRP-positive and BRP-depleted AZs. While our experiments in the BRP deficiency heterozygote showed that Cac levels are insensitive to ~35% reductions in BRP, the new and more thorough analysis of the BRP RNAi AZs reveals that even more significant reductions in BRP abundance still support normal levels of Cac at AZs. These data provide further support for the model that BRP’s presence or absence, rather than its abundance, dictates whether an AZ can accrue normal levels of Cac. In the BRP RNAi manipulations, two populations of AZs exist: those with some BRP and those without detectable levels (Figure 4 – Supplemental figure 1B). When Cac abundance is quantified at all Cac-positive AZs, a 30% drop in Cac AZ levels in the BRP RNAi is observed compared to controls (Figure 4 – Supplemental figure 1 C). When BRP-negative AZs are removed from the analysis (Figure 4 – Supplemental figure 1 D,E), the remaining BRP-positive AZ population displays normal Cac levels (Figure 4 – Supplemental figure 1 F,G). This result is striking, given that BRP abundance in these BRP-positive AZs in BRP RNAi NMJs is only ~20% of wildtype levels. This additional analysis strengthens our conclusions for this aspect of the manuscript.

“Besides the total abundance of scaffolding proteins assessed by NMJ immunostaining, relevant interactions with Cac could depend on a post-translational modification which would make both the levels and modification state relevant for Cac accumulation. This should be discussed.”

We fully agree that with the experiments presented here, our data only support a model that the abundance of the BRP protein is not rate-limiting for Cac accumulation at AZs. We cannot rule out a model where a subset of BRP proteins at the AZ are post-translationally modified, and that this subset could potentially rate-limit stabilizing interactions with Cac. We added a sentence acknowledging this possibility into our discussion.

4. “Please correct and provide further details on the tissue processing for Western blot analysis.”

We re-made the blot image so that the lower band is not cut off (Figure 4C). We also separated the BRP deficiency and BRP RNAi experiments into two representative panels and showed a control for each since they required slightly different controls (CRISPR Cac-GFP for the BRP deficiency and Cac-GFP recombined with C155 for the BRP RNAi). We also added additional information on tissue processing to the methods section as requested.

5. “An interesting observation the authors should follow up is that it appears that not all AZs are equally affected by reducing alpha2delta levels. From the examples shown it appears that particularly the BRP-rich AZs contain higher (near normal) Cac levels. Is that so?”

Indeed, as noted by the reviewer, the most mature AZs (which contain the highest BRP enrichment as shown in Akbergenova et al., 2018) are less affected by α 2-δ manipulations. As shown using timecourse analysis of α 2-δ phenotypes (Figure 5M,N), they either increase for the *α 2-δ* null mutation or appear in the third instar stage for deficiency heterozygote or following overexpression. AZs formed earlier that include the most mature subpopulation are less affected by these manipulations. However, unlike the BRP RNAi, where the highest Cac-positive AZs had normal Cac levels, AZs in the *α 2-δ* null mutant never reach wildtype Cac levels. In the histogram in Figure 5C showing the distribution of single AZ Cac levels across the AZ population, one can note that Cac levels in the *α 2-δ* mutant do not reach mature wildtype levels.

“The other interesting aspect is that sizable transmission remains even when NMJ immunostainings indicate a reduction of 64%. Considering the high cooperativity between ca^2+^ influx and neurotransmitter release, this is truly unexpected (assuming a dependence on the 4th power would predict a reduction by 99%). Similarly, over-expression increases the NMJ levels of Cac-GFP but not the evoked responses (the effect depicted on the 5th stimulus in Figure 5I/J is rather modest) and unexpectedly the short-term plasticity is unaffected. This could indicate that not all stained Cac-GFP proteins/voltage-gated ca^2+^ channels contribute to ca^2+^ influx. The manuscript could be further strengthened by more directly reading out ca^2+^ signals (e.g. using genetically encoded optical reporters or infused synthetic dyes) in these conditions. The authors should then relate the relative ca^2+^ influx to the Cac-GFP levels of the same NMJ to test whether indeed signals primarily depend on the NMJ levels of Cac or whether participation in ca^2+^ influx can be differentially regulated by another layer.”

For single action potentials, the highest-releasing AZs are predicted to drive a majority of release at the ca^2+^ concentrations tested. Because the most mature AZs are less affected by *α 2-δ* mutations, the % reduction in the top subset of AZs is less than the total reduction in average AZ Cac. We’ve included this quantification in Author response image 2 – you can see that if we restrict analysis to the top 30 AZs per NMJ, the *α 2-δ* mutant only has a ~50% decrease in Cac abundance per AZ. This would partially explain why the impact on release is less than might be expected. While we did not measure mini amplitude in our recordings of the *α 2-δ* mutants, it is possible that increased quantal size could also contribute to the increase in eEJC amplitude observed. The possibility that changes in ca^2+^ conductance per channel could also be at play is very exciting, and will be a topic for future work. We generated an endogenously CRIPSR-tagged ratiometric Cac-mRuby-GCaMP7s in an effort to measure the amount of ca^2+^ influx per channel, but unfortunately the GCaMP signal was not bright enough to use.

6. “If axonal delivery played a role, effects may be augmented for more distant body segments, did the authors test anything along those lines?”

To address the reviewer’s comment, we tested whether the α 2-δ Cac-depletion phenotype was modulated by axon length and found the amount of reduction in Cac per AZ is equal at segment 2 versus segment 6. These data have been added to Figure 5- – Supplemental figure 1. The data presented here cannot distinguish between a role for α 2-δ in promoting pre-axonal trafficking of Cac through the biosynthetic pathway versus a role in forward trafficking down the axon. The finding that axon length does not alter the α 2-δ phenotype is consistent with α 2-δ playing a role in forward trafficking prior to long-range axon transport, however this conclusion requires further study.

Reviewer 2:

Reviewer 2 described the work as a “highly significant contribution to our understanding of synapse assembly and VGCC regulation… that will have a lasting impact on the field of Cellular and molecular neuroscience.” Thank you for the kind comments! The reviewer requested several clarifications. The first comment pertaining to the flipstop timecourse is addressed above, while other points are discussed below.

2. Is Cac required for apposition?

This is a very interesting question that we did not address in the original study. To address this question, we analyzed apposition by measuring the percent of GluR fields without BRP puncta, and conversely the percent of BRP puncta lacking GluR fields. There was no change in apposition for either measurement in Cac Flpstop animals (Figure 1 – Supplemental figure 1 I-K).

3. Quantifying Cac overexpression levels: “They show by Western blot that loss of one copy of Cac results in roughly 50% Cac protein. They do experiments with Cac OE, but do not demonstrate that these MNs actually express more Cac protein by Western blot as they did for the heterozygotes”.

Whole-head western blot quantification of Cac levels in Cac deficiency heterozygotes is a good measurement for whole-neuron Cac levels given every cell was equally manipulated by this genomic change in Cac copy number. However, for elav-GAL4 driven Cac-GFP overexpression, whole-brain quantification of Cac levels is a less reliable metric for Ib motor neuron overexpression. Elav-GAL4 drives transgene expression in some glia, as well as to different extents in distinct neuronal subtypes. A better estimate of the fold-increase in Cac overexpression in Ib neurons can be calculated from the comparison between green endogenous channel level with and without red Cac-RFP overexpression by the M1-Ib GAL4 driver (Figure 3D, E). Green Cac-GFP levels are about 3-fold reduced at AZs following Cac-RFP overexpression, indicating that unless Cac-GFP and Cac-RFP have different trafficking properties that would be unexpected, the overexpression is likely ~4-fold greater than endogenous levels.

4. “Is α 2-δ overexpression sufficient to seed AZs?”

Α 2-δ overexpression does not drive additional AZ formation. Indeed, fewer AZs in α 2-δ overexpression animals are present compared to controls. This difference may reflect a faster growth rate of these animals, allowing fewer days for AZs to form prior to the third instar stage. Since we size-matched rather than age-matched our larvae in this experiment, and since AZ number is highly dependent on the animal’s age, we did not report AZ number in this study.

5. “I think it would strengthen the authors' argument that a2d is rate-limiting for Cac removal if they could show increased delivery/recycling when overexpressing a2d*”.*

Unfortunately, we were unable to measure recycling in the α 2-δ overexpression for technical reasons. First, the UAS-α 2-δ overexpression construct has a bright red marker in the locus that cannot be recombined out. This marker is bright enough in axons to generate a red glow over nearby NMJs that precludes reliable quantification of red Cac-maple signals, which is dimmer relative to this marker. Second, α 2-δ overexpression only produces a strong phenotype in the third instar stage, but the Cac-Maple experiments are focused on the earliest formed AZs which are present in the first instar stage of development. Therefore, we would not expect to see a difference in retention upon α 2-δ overexpression in this subset of AZs.

Reviewer 3:

Reviewer 3 described the work as a “an excellent foundation to unlock how CaV2 channel regulation at AZs is modified by such processes as neuronal activity and synaptic plasticity” and thought the work would be “of significant interest and importance to the field”. We thank the reviewer for his positive comments. The reviewer requested several clarifications and suggested a host of exciting and insightful future experimental directions, many of which are rich enough in scope that they will form independent topics for future studies in the lab. The first comment pertaining to the flipstop timecourse is addressed above in the shared comments section. Other points are discussed below.

*2. “Cac splice isoforms”:* As noted in the response to Reviewer 2, we found that endogenous Cac-GFP levels are about 3-fold reduced at AZs following UAS-Cac-RFP overexpression. How specific endogenously spliced Cac channels individually accumulate at NMJ AZs is beyond the scope of the current study.

A. Can the authors clarify whether their endogenous C-terminal Cac, presumably like the endogenous N-term Cac, is predicted to tag every predicted isoform?

Flybase predicts 18 Cac isoforms – 17 of those would be tagged by our C-terminal CRISPR approach, as only one is predicted to have a longer C-terminal tail. It is likely that we are in fact tagging all Cac isoforms with our method, as the one predicted C-terminal splice variant (Cac-RU) appears to be a sequencing error in Flybase that extends the C-terminus by read-through of the stop codon.

B. Were any functional (electrophysiological) differences found between wild type (all Cac isoforms and/or Cac N-term tag) vs Cac-C-term vs UAS-Cac (single isoform)? This would allow the authors to make a point about to what extent different Cac splice isoforms might contribute significant differences to Pr/diversity, at least as assessed through electrophysiological recordings.

This is an interesting point to follow up on in future studies of Cac structure-function relationships and the role of individual Cac isoforms. We did not address this in the current study.

C. Similarly, were different splice isoforms apparent on the anti-GFP blot?

There are a couple of large MW bands recognized with the anti-GFP blot in Cac-GFP transgenic animals, but it is unknown if this represents degradation or expression of unique splicing isoforms. This would require isoform-specific antibodies or CRISPR labeling of unique exons that is beyond the scope of the current study. The N- and C-terminal GFP labeled Cac CRISPR lines should in theory label all Cac isoforms, as all the predicted spicing occurs in internal exons except for the one C-terminal version that likely represents a sequencing error.

D. Previous studies have overexpressed Cac and saw no major changes in synaptic strength, already suggesting that Cac abundance at AZs is limited through post-transcriptional mechanisms. This point should be mentioned and the appropriate papers cited. The authors also use the vague term "biosynthesis", rather than a more clear terminology of "transcription" vs "translation" vs "post-translational mechanisms". If the authors use biosynthesis to mean the combined product of transcription, translation, and post-translational regulation, the authors should clearly define this in the manuscript.

As noted by the reviewer, work in mammalian systems also suggests post-transcriptional regulation – we have cited this literature in our manuscript in relation to the prior work on “slot” models for CaV2 channels at AZs, and added several new citations describing more recent studies in this area. Since we cannot separate regulation of transcription, mRNA stability or protein translation for several of our experimental manipulations, we used the term biosynthesis to encompass the idea that control could be occurring at several steps. We expect the majority of AZ Cac regulation will be post-translational. It will be exciting to follow up on this regulation in the future, particularly in relation to axonal transport versus AZ capture and retention. Using the Cac-maple toolkit, we have begun to address this for several of the manipulations described at the end of our manuscript, and it will form an important tool for us moving forward as we dissect this regulation in finer detail.

3. Other pathways/targets that may regulate Cac abundance/dynamics at AZs: The first half of this manuscript nicely confirms and extends much of what was known or expected about the importance of Cac in AZ assembly, and how BRP and α-2 δ relate to Cac. There is a lot of important data in this manuscript, and a reasonable point could be made that additional studies might be outside the scope of the current manuscript. But I would suggest the authors consider the following experiments to extend the importance and impact of this study:A. Can the authors test the role of the ca^2+^ channel Β subunit in Cac regulation?

This is an interesting direction to go in the future and we are planning on making genetic toolkits similar to Cac to test the role of this subunit in future studies.

B. Does Cac delivery or turnover change in conditions of hyper- or hypo-activity? There are numerous ways to manipulate activity in *Drosophila* neurons, and to what extent Cac abundance and turnover are sensitive to activity would add an intriguing dimension to this study.

We will be testing the role of activity, as well as single channel conductance, on Cac AZ targeting and turnover at AZs in the future. This will be interesting to compare to the literature on GluR trafficking in LTP and LTD from mammalian studies.

C. Are there any ideas or possible pathways of how a surplus pool of Cac is then limited and selectively trafficked to be incorporated at AZs? This bottleneck appears to be the key node of regulation, and while α-2 δ seems to be involved, some insight into the motors or transport vesicles would be of major interest.

How Cac AZ levels are mechanistically controlled through specific signaling pathways is a major interest for many in the field, and we hope to be able to contribute to these pathways with future genetic screening for mutants with defective Cac trafficking. Some of this likely involves folding and processing of the protein within the ER/Golgi network, while other points of regulation will likely involve cargo loading/unloading from molecular motors during axonal transport, as well as capture at individual AZs once the cargo has been released at the terminal. Whether this involves direct fusion of post-Golgi Cac containing vesicles directly at the AZ membrane, or rather fusion of these vesicles somewhere along the axonal plasma membrane and subsequent capture by active zone proteins is unknown.